# Assessment of hemagglutinin-inhibition activity following influenza vaccination during the 2022–2023, 2023–2024, and 2024–2025 seasons

Engin Berber[1¤], Hannah B. Hanley[2], Brianna M. Gamez[3], Ted M. Ross[1,2,3,4]*

1 Department of Infection Biology, Cleveland Clinic Research, Cleveland, Ohio, United States of America,
2 Center for Vaccines and Immunology, University of Georgia, Athens, Georgia, United States of America,
3 Florida Research and Innovation Center, Cleveland Clinic, Port Saint Lucie, Florida, United States of America, 4 Department of Infectious Diseases, University of Georgia, Athens, Georgia, United States of America

¤ Current address: Pathobiological Sciences, School of Veterinary Medicine, Louisiana State University, Baton Rouge, Louisiana, United States of America
* rosst7@ccf.org

## Abstract

Seasonal influenza vaccines are available in multiple formulations including egg-based inactivated standard-dose, cell-based inactivated, recombinant, and live-attenuated. However, vaccine effectiveness and antibody responses vary by vaccine type, recipient age, and prior influenza exposures. Both age and repeated annual vaccination can modulate immune outcomes through antigenic differences between historical and updated vaccine formulations. In this three-season (2022–2023–2024–2025) study, 1,328 participants aged 9–89 years received one of six vaccine types (Fluzone standard dose (SD) or high dose (HD), Fluad, Flucelvax, Flublok, Flumist). Collected serum samples were tested for hemagglutination inhibition (HAI) activity against vaccine strains and historical circulating strains before vaccination and day 28 post-vaccination. Seroprotection and seroconversion rates were analyzed by age group and vaccine platform. Repeated vaccination across consecutive seasons was associated with reduced HAI boosting to current vaccine strains in some participants, although vaccination broadly increased antibody titers to prior strains (back-boosting) and maintained high seroprotection rates from year to year. Participants ≥65 years had lower post-vaccination HAI titers compared to younger adults and had rapid waning of HAI titers than vaccinated younger adults, yet elderly participants with HAI titers <1:40 had higher seroconversion rates following vaccination, which may in part reflect lower baseline titers in this group allowing a ≥4-fold rise to meet the seroconversion definition with Fluzone HD or Fluad adjuvanted vaccines compared to younger vaccinated participants. Over the three seasons, younger adult participants vaccinated with Flublok had higher HAI titers than participants vaccinated with Fluzone SD against H1N1 and H3N2 strains. In contrast, post-vaccination HAI titers in elderly participants were comparable between Fluzone HD and Fluad recipients for

**Data availability statement:** All relevant data are within the manuscript and its Supporting information files.

**Funding:** This project has been funded as part of the Collaborative Influenza Vaccine Innovations Centers (CIVICs) by the National Institute of Allergy and Infectious Diseases, a component of the NIH, Department of Health and Human Services, under contract 75N93019C00052. T.M.R is also supported in part as a Georgia Eminent Scholar by the Georgia Research Alliance, GRA-001. The funders had no role in study design, data collection and analysis, decision to publish, or preparation of the manuscript.

**Competing interests:** The authors have declared that no competing interests exist.

each vaccine component, including influenza A and B strains. Participants vaccinated with Flumist had low or no measurable serum HAI activity. Overall, the elicitation of vaccine induced immune responses is strongly influenced by vaccine formulation, age and immune history of the participant.

## Introduction

Seasonal influenza viruses cause significant morbidity and mortality worldwide [1,2]. Vaccination remains the primary strategy for prevention of influenza virus infection and disease [3,4]. Annual epidemics are responsible for an estimated 3–5 million cases of severe illness and 290,000–650,000 deaths worldwide [2]. However, the effectiveness of influenza vaccines varies considerably from season to season due to continual antigenic evolution of the virus and therefore vaccines are updated annually [5]. Host factors, such as age and immune history, influence vaccine performance [6,7]. Older adults are disproportionately vulnerable to influenza virus complications and generally have lower vaccine-induced immune responses than standard-dose vaccinated younger adults [8,9].

Currently in the U.S., several seasonal influenza vaccine formulations are available: egg-based inactivated vaccines administered as standard-dose or high-dose (Fluzone, Sanofi Pasteur); squalene-based oil-in-water adjuvant (MF59) adjuvanted inactivated vaccine (Fluad, Sequiris); cell-culture–based inactivated vaccine (Flucelvax, Sequiris); recombinant hemagglutinin (HA) vaccine (Flublok, Sanofi Pasteur); and the live-attenuated intranasal vaccine (Flumist, AstraZeneca). HAI antibody titers are widely used as a correlate of protection and a surrogate marker of vaccine effectiveness [10,11]. Influenza vaccine elicited HAI immune responses can be compromised by repeated annual vaccination [12–15]. However, the impact of prior vaccination or comparing vaccine platforms have often been limited to a single season, vaccine formulation, or specific populations.

The goal of this study was to elucidate age-specific immune responses following vaccination with different vaccine formulations to determine the effects of prior vaccination on antibody outcomes over consecutive seasons. A comprehensive three-season study (2022–2023–2024–2025) was conducted to evaluate HAI antibody responses following influenza vaccination expressed from different vaccine platforms in participants from 9–89 years of age. A total of 1,328 participants were enrolled, 203 of whom received influenza vaccine in all three seasons forming a 'repeater' cohort for longitudinal assessment. Results were compared to participants who were not vaccinated in all three seasons. This observational study was designed to address four primary questions: (i) how baseline and post-vaccination HAI responses vary by age group across consecutive influenza seasons; (ii) within indicated populations, how licensed vaccine formulations compare in measured HAI responses; (iii) how baseline (pre-vaccination) titers influence observed post-vaccination fold-rise and seroconversion metrics; and (iv) how prior-season vaccination history and repeat vaccination patterns are associated with baseline titers and post-vaccination responses in a longitudinal repeater cohort.

## Materials and methods

### Study design and participants and vaccination

This observational (non-randomized) study was conducted over three consecutive influenza seasons between 2022–2023, 2023–2024, and 2024–2025 with vaccinations administered each year starting from September. A total of 1,328 volunteers aged 9–89 years were recruited from community-based clinics and university-affiliated sites, including the University of Georgia in Athens, GA, and Cleveland Clinic Florida in Port Saint Lucie, FL USA (Tables 1, S2). Enrollment criteria required participants to sign informed consent and to be generally healthy or to have stable chronic medical conditions, while exclusion criteria included acute febrile illness at the time of vaccination, concurrent participation to another influenza vaccine studies or contraindications to licensed influenza vaccines.

Participants were immunized with seasonally licensed influenza vaccines in accordance with standard clinical practice and age-based recommendations. Quadrivalent vaccine formulations contained the four influenza strains designated annually by the U.S. Food and Drug Administration (FDA) and based on the World Health Organization (WHO) recommendations for the Northern Hemisphere (Table 2).

For the 2024–2025 season (UGA9), however, vaccination followed the revised FDA guidance recommending a trivalent formulation, in which the B/Yamagata lineage was excluded. The vaccines administered across the study included high-dose egg-based inactivated influenza vaccine (Fluzone High-Dose; FZ HD; Sanofi-Pasteur, Swiftwater, PA, USA), standard-dose egg-based inactivated vaccine (Fluzone Standard Dose; FZ SD Sanofi-Pasteur, Swiftwater, PA, USA), MF59-adjuvanted inactivated vaccine (Fluad; FA, Seqirus, Holly Springs, NC, USA), cell culture–derived inactivated vaccine (Flucelvax; FC, Seqirus, Holly Springs, NC, USA), recombinant HA vaccine (Flublok; FB; Sanofi-Pasteur, Swiftwater, PA, USA), and live-attenuated influenza vaccine (Flumist; FM; Astra-Zeneca, Gaithersburg, MD, USA). In the UGA7 season (2022–2023), 378 participants received one of these five vaccines, FB, FC, FM, FZ HD, or FZ SD. During the UGA8 season (2023–2024), 460 individuals were vaccinated with one of these five vaccines, FA, FC, FM, FZ HD, or FZ SD. In the UGA9 season (2024–2025), 490 participants were immunized with one of these six vaccines, FA, FB, FC, FM, FZ HD, or FZ SD. Across all three seasons, a subset of individuals consistently returned for vaccination and blood sampling, forming the longitudinal "repeater cohort" (Tables 3 and S3).

### Informed consent and ethical approval

Written informed consent was obtained from all adult participants prior to enrollment. For individuals younger than 18 years, parental or legal guardian consent was required, and written assent was obtained from minors when appropriate. The study procedures included the collection of demographic information, as well as blood serum samples. All procedures were reviewed and approved by the WIRB–Copernicus Group Institutional Review Board (WCG IRB# 20224877) and by the University of Georgia Institutional Review Board. To protect confidentiality, all participants and their corresponding samples were de-identified and assigned alphanumeric codes. The study was conducted in full compliance with the ethical principles outlined in the Declaration of Helsinki and other applicable regulations governing research involving human participants.

### Sample collection and processing

At each study visit, approximately 70–90 mL of blood was collected per participant at three time points: on the day of vaccination (D0), day 7 (D7), and again at day 28 (D28) following vaccination. Blood specimens were processed to isolate serum and samples were preserved at –30 °C (±10 °C). For this study, analyses focused on sera collected at D0 and D28, which were assessed for HAI activity against panels of influenza A/H1N1, A/H3N2, and influenza B viruses. These panels included both historical and current vaccine strains, as designated by the WHO for seasonal influenza vaccines.

**Table 1. Demographics of participants by age and vaccine type across the 2022–2023 to 2024–2025 influenza seasons.**

| Season (Cohort ID) | Age Groups and vaccine types | Total, n[a] | Average Age (Min-Max) | Gender (%) | | | Race/Ethnicity (%) | | | | | | Body Mass Index (BMI)[e] (%) | | | |
|---|---|---|---|---|---|---|---|---|---|---|---|---|---|---|---|---|
| | | | | Female | Male | Other[b] | White | Black or AA | Hispanic or Latino | Asian | Mixed[c] | Other[d] | < 25 | 25–30 | ≥ 30 | N/A |
| 2022-2023 (UGA7) | ALL | 378 | 47.1 (10-87) | 65.3% | 33.6% | 1.1% | 82.3% | 8.7% | 5.8% | 1.1% | 0.3% | 1.9% | 36.8% | 31.2% | 29.6% | 2.4% |
| | 9-17 | 36 | 14.2 (10-17) | 5.3% | 3.7% | 0.5% | 6.1% | 2.1% | 0.8% | 0.0% | 0.0% | 0.5% | 5.8% | 1.9% | 1.6% | 0.3% |
| | 18–34 | 85 | 26.8 (18-34) | 16.9% | 5.3% | 0.3% | 17.5% | 1.3% | 1.9% | 0.8% | 0.0% | 1.1% | 10.8% | 6.9% | 4.2% | 0.5% |
| | 35–49 | 82 | 42.0 (35-49) | 14.3% | 7.4% | 0.0% | 16.7% | 2.6% | 2.4% | 0.0% | 0.0% | 0.0% | 4.8% | 7.7% | 7.9% | 1.3% |
| | 50–64 | 73 | 57.1 (50-64) | 14.0% | 5.0% | 0.3% | 16.7% | 1.9% | 0.5% | 0.0% | 0.0% | 0.3% | 5.6% | 5.6% | 7.9% | 0.3% |
| | ≥65 | 102 | 72.6 (65-87) | 14.8% | 12.2% | 0.0% | 25.4% | 0.8% | 0.3% | 0.3% | 0.3% | 0.0% | 9.8% | 9.3% | 7.9% | 0.0% |
| | Fluad | 0 | 0 | 0.0% | 0.0% | 0.0% | 0.0% | 0.0% | 0.0% | 0.0% | 0.0% | 0.0% | 0.0% | 0.0% | 0.0% | 0.0% |
| | Flublok | 44 | 42.8 (18-78) | 68.2% | 31.8% | 0.0% | 84.1% | 9.1% | 6.8% | 0.0% | 0.0% | 0.0% | 3.7% | 5.6% | 2.1% | 0.3% |
| | Flucelvax | 44 | 36.5 (11-63) | 77.3% | 20.5% | 2.3% | 54.5% | 25.0% | 11.4% | 4.5% | 0.0% | 4.5% | 4.5% | 2.9% | 3.7% | 0.5% |
| | Flumist | 45 | 30.9 (14-49) | 71.1% | 24.4% | 4.4% | 66.7% | 15.6% | 15.6% | 2.2% | 0.0% | 0.0% | 5.3% | 3.7% | 2.9% | 0.0% |
| | Fluzone HD | 90 | 72.4 (65-87) | 54.4% | 45.6% | 0.0% | 95.6% | 1.1% | 1.1% | 1.1% | 1.1% | 0.0% | 9.3% | 7.4% | 7.1% | 0.0% |
| | Fluzone SD | 155 | 41.3 (10-84) | 65.8% | 33.5% | 0.6% | 86.5% | 6.5% | 3.9% | 0.0% | 0.0% | 3.2% | 14.0% | 11.6% | 13.8% | 1.6% |
| 2023-2024 (UGA8) | ALL | 460 | 49.1 (9-88) | 65.7% | 33.7% | 0.7% | 83.5% | 7.2% | 5.2% | 3.3% | 0.9% | 0.0% | 34.3% | 34.3% | 30.9% | 0.4% |
| | 9-17 | 30 | 13.4 (9-17) | 2.6% | 3.7% | 0.2% | 4.3% | 1.3% | 0.2% | 0.0% | 0.7% | 0.0% | 4.1% | 0.9% | 1.1% | 0.4% |
| | 18–34 | 106 | 27.2 (18-34) | 16.1% | 6.7% | 0.2% | 17.2% | 1.7% | 2.0% | 2.0% | 0.2% | 0.0% | 9.6% | 8.9% | 4.6% | 0.0% |
| | 35–49 | 92 | 41.5 (35-49) | 14.6% | 5.4% | 0.0% | 15.7% | 2.0% | 1.7% | 0.7% | 0.0% | 0.0% | 4.6% | 6.7% | 8.7% | 0.0% |
| | 50–64 | 89 | 56.3 (50-64) | 14.6% | 4.6% | 0.2% | 17.2% | 1.1% | 0.7% | 0.4% | 0.0% | 0.0% | 4.8% | 6.3% | 8.3% | 0.0% |
| | ≥65 | 143 | 73.2 (65-88) | 17.8% | 13.3% | 0.0% | 29.1% | 1.1% | 0.7% | 0.2% | 0.0% | 0.0% | 11.3% | 11.5% | 8.3% | 0.0% |
| | Fluad | 52 | 73.2 (65-86) | 59.6% | 40.4% | 0.0% | 92.3% | 3.8% | 3.8% | 0.0% | 0.0% | 0.0% | 3.7% | 4.6% | 3.0% | 0.0% |
| | Flublok | 0 | 0 | 0.0% | 0.0% | 0.0% | 0.0% | 0.0% | 0.0% | 0.0% | 0.0% | 0.0% | 0.0% | 0.0% | 0.0% | 0.0% |
| | Flucelvax | 121 | 39.1 (9-78) | 71.9% | 27.3% | 0.8% | 77.7% | 10.7% | 9.9% | 1.7% | 0.0% | 0.0% | 7.2% | 10.4% | 8.7% | 0.0% |
| | Flumist | 46 | 28.4 (9-48) | 71.7% | 23.9% | 4.3% | 67.4% | 10.9% | 15.2% | 2.2% | 4.3% | 0.0% | 4.1% | 3.3% | 2.4% | 0.2% |
| | Fluzone HD | 83 | 73.3 (65-88) | 57.8% | 42.2% | 0.0% | 97.6% | 0.0% | 1.2% | 1.2% | 0.0% | 0.0% | 7.2% | 5.9% | 5.0% | 0.0% |
| | Fluzone SD | 158 | 42.0 (9-78) | 65.2% | 34.8% | 0.0% | 82.3% | 8.2% | 1.3% | 7.0% | 1.3% | 0.0% | 12.2% | 10.2% | 11.7% | 0.2% |

*(Continued)*

**Table 1.** (Continued)

| Season (Cohort ID) | Age Groups and vaccine types | Total, nᵃ | Average Age (Min-Max) | Gender (%) | | | Race/Ethnicity (%) | | | | | | Body Mass Index (BMI)ᵉ (%) | | | |
|---|---|---|---|---|---|---|---|---|---|---|---|---|---|---|---|---|
| | | | | Female | Male | Otherᵇ | White | Black or AA | Hispanic or Latino | Asian | Mixedᶜ | Otherᵈ | < 25 | 25–30 | ≥ 30 | N/A |
| 2024-2025 (UGA9) | ALL | 490 | 52.4 (9-89) | 65.3% | 34.3% | 0.4% | 82.9% | 8.4% | 6.7% | 1.6% | 0.0% | 0.4% | 31.6% | 36.7% | 31.6% | 0.0% |
| | 9-17 | 26 | 13.5 (9-17) | 2.2% | 2.9% | 0.2% | 2.9% | 1.8% | 0.6% | 0.0% | 0.0% | 0.0% | 3.3% | 1.2% | 0.8% | 0.0% |
| | 18–34 | 90 | 27.3 (18-34) | 12.4% | 5.9% | 0.0% | 13.7% | 1.6% | 2.0% | 1.0% | 0.0% | 0.0% | 6.1% | 6.1% | 6.1% | 0.0% |
| | 35–49 | 105 | 41.9 (35-49) | 15.3% | 6.1% | 0.0% | 16.3% | 2.2% | 2.4% | 0.4% | 0.0% | 0.0% | 5.1% | 8.4% | 8.0% | 0.0% |
| | 50–64 | 88 | 57.3 (50-64) | 13.1% | 4.7% | 0.2% | 15.3% | 1.4% | 0.6% | 0.2% | 0.0% | 0.4% | 4.7% | 5.5% | 7.8% | 0.0% |
| | ≥65 | 181 | 74.1 (65-89) | 22.2% | 14.7% | 0.0% | 34.7% | 1.2% | 1.0% | 0.0% | 0.0% | 0.0% | 12.4% | 15.5% | 9.0% | 0.0% |
| | Fluad | 76 | 74.1 (66-87) | 63.2% | 36.8% | 0.0% | 93.4% | 3.9% | 2.6% | 0.0% | 0.0% | 0.0% | 4.9% | 6.7% | 3.9% | 0.0% |
| | Flublok | 58 | 44.7 (18-79) | 72.4% | 27.6% | 0.0% | 77.6% | 10.3% | 10.3% | 1.7% | 0.0% | 0.0% | 3.1% | 4.3% | 4.5% | 0.0% |
| | Flucelvax | 81 | 39.4 (9-74) | 71.6% | 27.2% | 1.2% | 71.6% | 12.3% | 11.1% | 3.7% | 0.0% | 1.2% | 4.9% | 6.1% | 5.5% | 0.0% |
| | Flumist | 38 | 30.7 (10-49) | 65.8% | 31.6% | 2.6% | 60.5% | 15.8% | 23.7% | 0.0% | 0.0% | 0.0% | 3.3% | 2.4% | 2.0% | 0.0% |
| | Fluzone HD | 97 | 74.4 (65-89) | 57.7% | 42.3% | 0.0% | 95.9% | 1.0% | 3.1% | 0.0% | 0.0% | 0.0% | 7.1% | 8.0% | 4.7% | 0.0% |
| | Fluzone SD | 140 | 41.9 (10-65) | 65.0% | 35.0% | 0.0% | 82.9% | 10.7% | 2.9% | 2.9% | 0.0% | 0.7% | 8.4% | 9.2% | 11.0% | 0.0% |

This table describes the overall study population. Percentages are calculated within each row. Age is reported in years as mean (min–max). Gender and Race/ethnicity categories reflect self-report. Abbreviations: SD, Standard Dose; HD, High-Dose; AA, African American.

ᵃ Participants are counted once within each season (UGA7 n = 378; UGA8 n = 460; UGA9 n = 490). Longitudinal repeaters may contribute up to three seasons.

ᵇ "Other" gender includes participants not identifying as female or male.

ᶜ "Mixed" indicates multiracial self-identification (self-reported).

ᵈ "Other" race/ethnicity indicates self-reported minorities not listed in race groups.

ᵉBMI categories are < 25, 25–30, and ≥30 kg/m²; N/A indicates missing BMI.

## Hemagglutination inhibition (HAI) assay

Hemagglutination inhibition (HAI) assays were conducted to measure serum antibody responses against influenza HA, specifically those capable of preventing virus-mediated agglutination of red blood cells (RBCs). The procedure was adapted from the WHO influenza virological surveillance manual with modifications tailored to this study. Prior to analysis, serum samples were pretreated with receptor-destroying enzyme (RDE; Denka Seiken, Co., Japan) to eliminate nonspecific inhibitors. Three parts RDE were added to one part serum and incubated overnight at 37°C. The following day, residual enzyme activity was neutralized by heating the samples at 56°C for 30–60 min, after which they were diluted tenfold with phosphate-buffered saline (PBS). RDE-treated sera were serially diluted twofold in 96-well V-bottom microtiter plates (Thermo Fisher Scientific, Waltham, MA, USA). Each well was then mixed with an equal volume of influenza virus suspension standardized to 8 hemagglutination units (HAU) per 50 µL. After 20 min of incubation at room temperature (RT), RBCs were added: for most assays, 50 µL of 0.8% turkey RBCs (Lampire Biologicals,

**Table 2. Vaccine formulations used for Northern Hemisphere for three seasons (2022-2023, 2023-2024 and 2024-2025).**

| Season | Vaccine Type | IAV (H1N1)[a] | IAV (H3N2)[b] | IBV (Victoria) | IBV (Yamagata) |
|---|---|---|---|---|---|
| 2022–2023 | Egg-based (Fluzone HD/SD, FluMist) | A/Victoria/2570/2019 | A/Darwin/9/2021 | B/Austria/1359417/2021 | B/Phuket/3073/2013 |
| | Cell/Recombinant (Flucelvax, Flublok) | A/Wisconsin/588/2019 | A/Darwin/6/2021 | B/Austria/1359417/2021 | B/Phuket/3073/2013 |
| 2023–2024 | QIV Egg-based (Fluzone HD/SD, Fluad, FluMist) | A/Victoria/4897/2022 | A/Darwin/9/2021 | B/Austria/1359417/2021 | B/Phuket/3073/2013 |
| | Cell/Recombinant (Flucelvax, Flublok) | A/Wisconsin/67/2022 | A/Darwin/6/2021 | B/Austria/1359417/2021 | B/Phuket/3073/2013 |
| 2024–2025 | Egg-based (Fluzone HD/SD, Fluad, FluMist) | A/Victoria/4897/2022 | A/Thailand/8/2022[b] | B/Austria/1359417/2021 | Excluded[c] |
| | Cell/Recombinant (Flucelvax, Flublok) | A/Wisconsin/67/2022 | A/Massachu-setts/18/2022[b] | B/Austria/1359417/2021 | Excluded[c] |

Table showing vaccine-component strains by platform (egg-based vs. cell/recombinant) used across three seasons.

[a]IAV (H1N1) strains are selected to reflect circulating viruses, with differences based on vaccine platform (egg-adapted vs. cell/recombinant propagated strains).

[b]A/Thailand/8/2022 was selected for egg-based vaccines, while A/Massachusetts/18/2022 was chosen for cell/recombinant vaccines in 2024–2025.

[c]For 2024–2025, B/Yamagata was excluded from vaccine formulations by the WHO recommendation due to the apparent absence of circulating B/Yamagata viruses globally.

**Table 3. Demographics of repeater volunteers for three consecutive seasons (2022–2023 to 2024–2025).**

| Age groups | Total n (%) | Gender n (%)[a] | | Race/Ethnicity n (%) | | | | Body mass index (BMI) n (%)[b] | | | |
|---|---|---|---|---|---|---|---|---|---|---|---|
| | | Female | Male | White | Black or AA | Hispanic or Latino | Other[c] | <25 | 25-30 | ≥30 | N/A |
| All | 203 (100.0%) | 136 (67.0%) | 66 (32.5%) | 179 (88.2%) | 15 (7.4%) | 5 (2.5%) | 4 (2.0%) | 70 (34.5%) | 66 (32.5%) | 65 (32.0%) | 2 (1.0%) |
| 9-17 | 15 (7.4%) | 9 (4.4%) | 5 (2.5%) | 8 (3.9%) | 4 (2.0%) | 1 (0.5%) | 2 (1.0%) | 10 (4.9%) | 2 (1.0%) | 3 (1.5%) | 0 (0.0%) |
| 18-34 | 31 (15.3%) | 28 (13.8%) | 3 (1.5%) | 26 (12.8%) | 2 (1.0%) | 1 (0.5%) | 2 (1.0%) | 14 (6.9%) | 8 (3.9%) | 8 (3.9%) | 1 (0.5%) |
| 35-49 | 43 (21.2%) | 32 (15.8%) | 11 (5.4%) | 36 (17.7%) | 5 (2.5%) | 2 (1.0%) | 0 (0.0%) | 9 (4.4%) | 14 (6.9%) | 20 (9.9%) | 0 (0.0%) |
| 50-64 | 37 (18.2%) | 25 (12.3%) | 12 (5.9%) | 36 (17.7%) | 1 (0.5%) | 0 (0.0%) | 0 (0.0%) | 9 (4.4%) | 11 (5.4%) | 16 (7.9%) | 1 (0.5%) |
| ≥65 | 77 (37.9%) | 42 (20.7%) | 35 (17.2%) | 73 (36.0%) | 3 (1.5%) | 1 (0.5%) | 0 (0.0%) | 28 (13.8%) | 31 (15.3%) | 18 (8.9%) | 0 (0.0%) |

Longitudinal repeater cohort (n = 203), defined as participants with paired D0 and D28 sera available in all three seasons from 2022–2023–2024–2025 (UGA7–UGA9).

Age and BMI represent measurements at first enrollment (2022–2023 season) for the longitudinal repeater cohort. Percentages are calculated within the repeater cohort. Abbreviations: AA, African American; BMI: Body mass index.

[a] "Other" race/ethnicity indicates self-reported minorities not listed in race groups.

[b] BMI categories are <25, 25–30, and ≥30 kg/m²; N/A indicates missing BMI.

[c] Gender totals may not sum to 100% because one participant in the 9–17 years group reported a "other" response.

Pipersville, PA, USA) diluted and prepared in 1x fresh PBS. RBCs were washed twice with 1xPBS, stored at 4°C and used within 24 h of preparation. For H3N2 viruses, hemagglutination inhibition was performed using guinea pig RBCs at a final concentration of 0.75% (Lampire Biologicals) supplemented with 20 nM oseltamivir, a neuraminidase inhibitor, to enhance assay specificity. Plates were gently agitated, covered, and incubated for 30 min at RT prior to scoring. The HAI titer was defined as the reciprocal of the highest serum dilution that completely prevented hemagglutination. Positive control sera from ferret or mouse infections were included to ensure assay consistency across runs. For interpretation, HAI titer outcomes were classified relative to the seroprotection threshold (HAI titer ≥1:40). Titers

≥1:40 were categorized as at/above the seroprotection threshold and titers <1:40 as below the seroprotection threshold (non-seroprotective). Seroconversion was defined as ≥4-fold rise in HAI titer from baseline with a post-vaccination value ≥1:40, and seroprotection as any post-vaccination HAI titer ≥1:40. HAI titers <1:10 were treated as undetectable and were assigned a value of 1:5 for calculation purposes in accordance with guidelines from the European Committee for Proprietary Medicinal Products (CPMP) and the U.S. FDA [10,11].

### Viruses tested in HAI

The influenza viruses employed in this study were obtained through the Influenza Reagent Resource (IRR), Biodefense and Emerging Infections (BEI) Research Resources, or supplied by Sanofi Pasteur. To maintain consistency with egg-derived vaccine formulations, all viruses were propagated in 10-day-old embryonated chicken eggs following WHO guidelines. Virus stocks were passaged once in eggs, titrated with turkey RBCs, aliquoted into single-use vials, and stored at −80 °C until use.

For the H1N1 subtype, the panel included A/California/07/2009 (CA/09), A/Brisbane/02/2018 (BR/18), A/Guangdong-Maonan/SWL1536/2019 (GD/19), A/Victoria/2570/2019 (VC/19), and A/Victoria/4897/2022 (VC/22). The H3N2 panel comprised A/Hong Kong/4801/2014 (HK/14), A/Singapore/INFIMH-16–0019/2016 (SG/16), A/Kansas/14/2017 (KS/17), A/South Australia/34/2019 (SA/19), A/Hong Kong/2671/2019 (HK/19), A/Tasmania/503/2020 (TS/20), A/Darwin/9/2021 (DR/21), A/Massachusetts/2022 (MA/22), and A/District of Columbia/27/2023 (DC/23). For influenza B, the Yamagata-lineage strain was B/Phuket/3073/2013 (B/PH/13), while the Victoria-lineage strains included B/Colorado/06/2017 (B/CO/17), B/Washington/02/2019 (B/WA/19), and B/Austria/1359417/2021 (B/AS/21)

### Statistical analysis

All HAI titers were $\log_2$-transformed prior to statistical testing using the formula $y = log_2(y)$ to normalize data distribution. Tables were generated using Microsoft Excel (Microsoft 365, version 2506, Redmond, WA, US) with embedded formulas to ensure data validation and minimize transcription errors. Statistical analysis of HAI titers was conducted using GraphPad Prism (version 10, GraphPad Software, San Diego, CA, USA). Comparisons between baseline (D0) and post-vaccination (D28) titers within each vaccine group and between vaccine types at D28 were evaluated using one-way ANOVA with Games-Howell's multiple comparisons test. In addition, paired within-subject changes from D0 to D28 were assessed using the Wilcoxon matched-pairs signed-rank test. A p-value < 0.05 was considered statistically significant, with thresholds denoted as: $p < 0.05$ (*), $p < 0.01$ (**), $p < 0.001$ (***), and $p < 0.0001$ (****).

## Results

### Demographics of volunteers

From 2022–2023–2024–2025, a total of 1,328 participants aged 9–89 years were enrolled across three consecutive influenza seasons (Table 1). The average age per cohort ranged from 47.1 years during the 2022–2023 season to 52.4 years during the 2024–2025 season. Across all seasons, the cohorts were majority female (65.3–65.7%), predominately White (82.3–83.5%), and included representation from Black/African American (7.2–8.7%), Hispanic or Latino (5.2–6.7%), Asian (1.1–3.3%), and other racial/ethnic groups (<2%). Body mass index (BMI) distributions were relatively balanced across <25, 25–30, and ≥30 BMI categories, though a slight shift toward higher BMI was noted in the 2024–2025 cohort.

Age distribution varied between seasons, with the ≥65 years category representing 27.0% of participants in 2022–2023 season, 31.1% in 2023–2024 season, and 36.9% in 2024–2025 season. The proportion of adolescents (9–17 years) was relatively small across all three seasons compared to other age groups, decreasing from 9.5% in 2022–2023 to 6.5% in 2023–2024 and further to 5.3% in 2024–2025. A variety of vaccine platforms were administered each season, including high-dose egg-based (Fluzone HD), standard-dose egg-based (Fluzone SD), adjuvanted egg-based (Fluad), cell-based

(Flucelvax), recombinant HA (Flublok), and live attenuated (Flumist) formulations. All vaccines were administered according to licensed manufacturer age indications and FDA-approved, age-based recommendations.

## Vaccine formulations

The antigenic composition of vaccines varied between seasons and platforms (Table 2). In 2022–2023 season, the H1N1 component was VC/19-like for egg-based vaccines and A/Wisconsin/588/2019 (WI/19)-like for cell/recombinant vaccines. The H3N2 component was DR/21-like (egg-based) or DR/21-like (cell/recombinant-based). Both B lineages B/AS/21 Victoria-like and B/PH/13 Yamagata-like were included across platforms. In the 2023–2024 season, the H1N1 component was updated to VC/22-like (egg-based) or A/Wisconsin/67/2022 (WI/22)-like (cell/recombinant-based), while the H3N2 component remained DR/21-like (egg-based) or DR/21-like (cell/recombinant-based). The B components were unchanged from the 2022–2023 season. In the 2024–2025 season, the B/Yamagata lineage was excluded from WHO recommendations and all licensed vaccines and only B/AS/21-like (Victoria lineage) was included. The H3N2 component was updated to A/Thailand/8/2022 (TH/22)-like (egg-based) or A/Massachusetts/18/2022 (MA/22)-like (cell/recombinant-based). The H1N1 component remained VC/22-like (egg-based) or WI/22-like (cell/recombinant-based).

## Demographics of the three-season repeater cohort

A total of 203 individuals participated in each of the three study seasons and form the longitudinal repeater cohort (Tables 3, S1 and S3). This cohort was 67% female, 88.2% White, and had a mean age skewed toward older adults, with 37.9% aged ≥65 years. BMI distribution was balanced across <25 (34.5%), 25–30 (32.5%), and ≥30 (32%). FluMist was administered primarily to younger participants (overall age range 14–49 years; mean age ~32–34 years across seasons consistent with package insert (FluMist, AstraZeneca; 2–49 years).

## Vaccination patterns among repeaters

S1 Table summarizes vaccine-type sequences received by repeater participants across the three seasons (n = 203). The most common stable pattern was Fluzone SD in all three years (FZ SD > FZ SD > FZ SD; n = 59), followed by Fluzone HD in all three years (FZ HD > FZ HD > FZ HD; n = 54), Flumist in all three years (FM > FM > FM; n = 16), and Flucelvax in all three years (FC > FC > FC; n = 15).

## HAI antibody responses

Serum samples collected from participants on the day of vaccination (D0) and 28 days later (D28) in each season and were assessed by HAI against panels of historical and contemporary A/H1N1, A/H3N2, and influenza B viruses representing both vaccine and non-vaccine strains. Age-stratified heat maps (Fig 1 and S4 Table) illustrate baseline immunity, post-vaccination responses, and the persistence or waning of protective titers across seasons.

Participants 9–17 year of age had the highest D0 HAI titers against multiple H1N1 viruses with clear back-boosting to antigenically related earlier strains (Fig 1A). HAI titers were maintained against older isolates through the subsequent season's baseline, whereas HAI titers against most recent vaccine strains declined prior to the next vaccination's baseline. Participants 18–34 years old had baseline HAI titers lower than participants 9–17 years old in most seasons, but with fewer participants having HAI titers >1:40 prior to vaccination at D0. Following vaccination, antibody responses were directed to the vaccine components, as well as HAI titers against older H1N1 influenza viruses, although breadth and magnitude of the HAI titers were slightly reduced relative to participants 9–17 years of age. Participants 35–64 year of age had lower baseline HAI titers on D0. Following vaccination, there were strain-matched rises in HAI titers, but a more limited increase in cross-reactive HAI titers that had waned by the subsequent season's baseline. Adults ≥65 years of age consistently had the lowest HAI titers on D0 titers. Following vaccination, these participants had seroprotective HAI

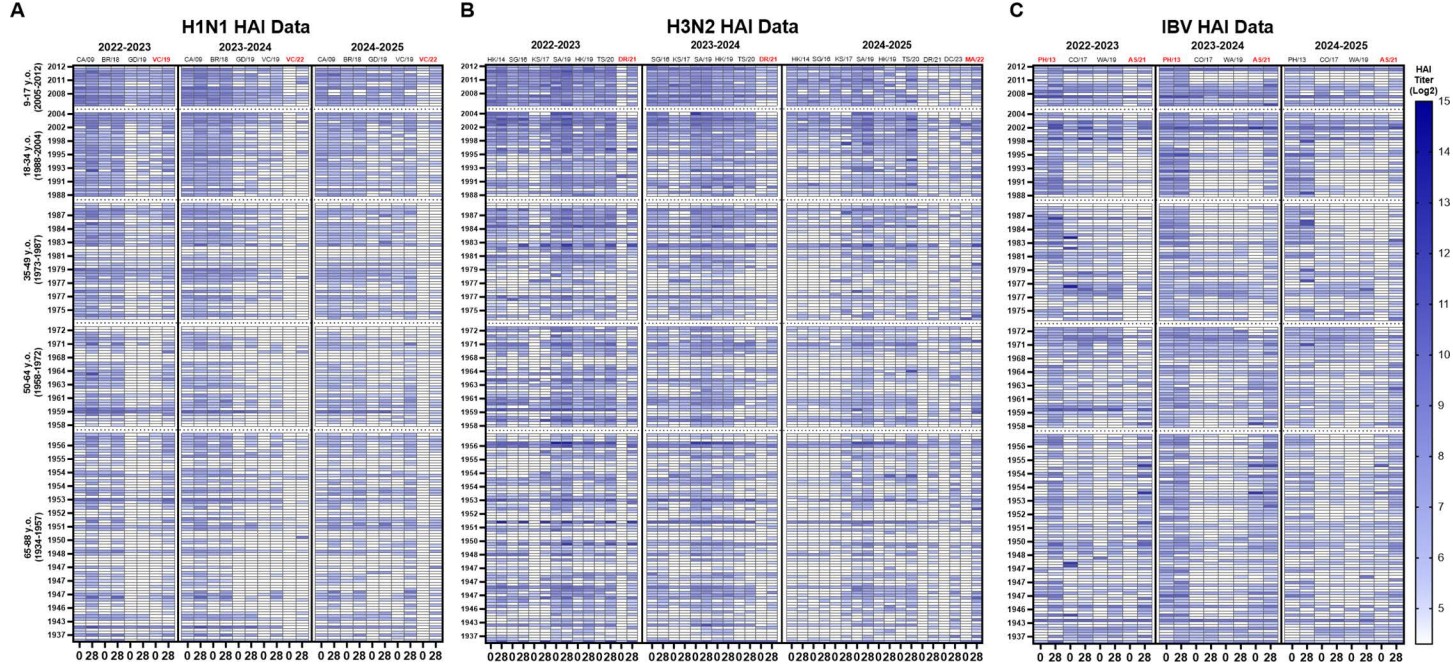

**Fig 1. Heat maps of hemagglutination inhibition (HAI) activity for each study season in repeater cohort.** Serum specimens obtained from longitudinal repeater cohort only (n = 203) prior to vaccination (D0) and 28 days after vaccination (D28) were evaluated against panels of historical influenza A and B viruses. Results are shown for (A) H1N1, (B) H3N2, and (C) influenza B viruses, with separate heat maps for the 2022–2023, 2023–2024, and 2024–2025 seasons. Viral antigens are ordered chronologically along on the x-axis from the earliest to the most recent isolates. Vaccine components are highlighted in red color on the top of x axis. Participants (rows) are grouped by birth year into defined age categories, ranging from the youngest at the top to the oldest at the bottom; within each age group. HAI titers are expressed as log$_2$ values. The vertical blue gradient bar on the far right indicates the color scale key (log$_2$ HAI titer values from 5 to 14) and applies to all panels. Titers ≥1:40 are displayed using a graded blue color scale with lighter shades indicating lower titers and darker shades indicating higher titers. Titers below 40 are indicated in white, reflecting non-seroprotective. Longitudinal repeater cohort (n = 203), defined as participants with paired D0 and D28 sera available in all three seasons and same individuals are shown across seasons. Within each strain, paired columns represent D0 (left) and D28 (right). Notably, the B/Yamagata lineage was not included in the 2024-2025 vaccine formulation; measured titers against Yamagata antigens in 2024-2025 reflect pre-existing immunity/back-boosting against B/PH/13 antigen rather than responses to a vaccine component.

titers of >1:40 against the H1N1 vaccine component of each season, but lower HAI titers against a more limited number of historical H1N1 influenza viruses.

HAI titers were uniformly lower and more heterogeneous at D0 against H3N2 viruses than HAI titers against the H1N1 influenza viruses (Fig 1B). Participants 9–17 years of age had highest baseline HAI titers against recent H3N2 isolates, but had the lowest HAI titers against the H3N2 vaccine component compared to other age groups. Following vaccination, there were increased HAI titers against the vaccine components for each season (*e.g.*, DR/21 in the 2022–2023 season and in the 2023–2024 season, MA/22 in the 2024–2025 season). Fewer participants 18–34 years old had HAI titers >1:40 at D0 during the 2022–2023 season compared with the subsequent two seasons. Following vaccination, HAI titers increased by D28 (>1:40), and modest back-boosting against historical H3N2 isolates was observed. For 35–64 year old participants, HAI titers on D0 were lower than younger participants with strain-specific increases in HAI titers following vaccination. Participants ≥65 years old had the lowest D0 HAI titers against the panel of H3N2 viruses compared to younger participants. Following vaccination, these HAI titers increased against each H3N2 vaccine per season, but immune responses waned prior to the subsequent season's baseline.

Participants between 9–17 years of age had detectable HAI titers at D0 against both influenza B virus (IBV) Yamagata and Victoria lineage components (Fig 1C). There was an increase in HAI titers following vaccination against the vaccine

components each season at D28 (*e.g.*, B/PH/13 for Yamagata when included and B/CO/17, B/WA/19, B/AS/21 for Victoria). HAI titers were also detected post-vaccination against IBV historical strains within the same lineage. Young adults (18–34 years of age) had HAI titers against B/Phu/13, but often lower HAI titers against newer B/Victoria influenza B viruses than elderly participants. Following vaccination, there was a rise in HAI titers against the B/Victoria lineage vaccine components with lower cross-lineage HAI activity. In adults 35–64 years old, there were low cross-lineage HAI titers at D0 and vaccination responses were more strain-specific and declined prior to next season. Participants ≥65 years of age had low to undetectable HAI titers against both lineages. Following vaccination, most elderly participants had HAI titers >1:40 against the B/Victoria vaccine component each season, but these antibody titers declined over the season and back to baseline by the start of the following influenza season. Notably, during the 2024–2025 season, the B/Yamagata lineage was not included in the vaccine and therefore B/Yamagata (*e.g.*, B/Phu/13) HAI titers represented historical antibody titers. There was a rise in HAI titers against the IBV Victoria component (e.g., B/WA/19, B/AS/21) in vaccinated participants during the 2024–2025 season.

**Average HAI titer responses by age group and season**

Over the three study seasons, the average hemagglutination inhibition (HAI) titers at baseline (D0) and post-vaccination (D28) varied by age group, vaccine component, and season (Fig 2 and S5 Table). HAI titers significantly increased against all four vaccine components across age groups following vaccination (Fig 3 and S5 Table). During the 2022–2023 season, participants between the ages of 9 and 17 years old had high pre-vaccination HAI titers against the H1N1 component VC/19 and the H3N2 component DR/21 with modest increases in HAI titers following vaccination at D28 (Figs 2A-B and 3A-B). In contrast, participants ≥65 years old had significantly lower titers against both H1N1 and H3N2 vaccine components compared to adolescents, but there were more pronounced fold-increases in post-vaccination HAI titers against the H1N1 and H3N2 vaccine components (Figs 2A-B and 3A-B). Participants between the ages of 18–34 and 35–49 years had HAI activity against the panel of influenza viruses although the titers progressively decreased with increasing age. During the 2023–2024 season, HAI titers at D0 were lower against the H1N1 but higher against H3N2 component of the vaccine across all age groups with the participants between the age of 18 and 34 years having the highest pre-vaccination D0 HAI titers. Following vaccination, fold-increases in HAI titers were modest across age groups for H1N1 and generally weaker in older participants, particularly people ≥65 years of age. During the 2024–2025 influenza season, participants aged 9–17 years had slightly higher D0 HAI titers against the H1N1 and the H3N2 components compared to other age groups, with modest fold-increases from baseline (post-vaccination HAI titers remained among the highest of all age groups). Participants aged 50–64 and ≥65 years had lower D0 HAI titers compared to younger participants (Figs 2A-B and 3A-B).

Participants between the ages of 9–17 years had among the highest D0 HAI titers against B/Yamagata in each season. During the 2022–2023 and the 2023–2024 seasons, these participants had HAI titers against the B/Yamagata lineage component that rose following vaccination at D28 (Figs 2C, 3C and 3G). During the 2024–2025 influenza season, participants 9–17 years of age had a statistically similar HAI titers between D0 and D28 (Figs 2C and 3K). During the 2022–2023 season, 9–17 years old participants had higher D0 HAI titers against the B/Victoria component than older participants with moderate to robust increases post-vaccination D28 HAI titers. Subsequently, there were increases in HAI titers against the B/Victoria component in participants during the 2023–2024 and 2024–2025 seasons (Figs 2D, 3L). Participants aged 18–34 years had high D28 HAI titers against the B/Victoria component following vaccination with the largest rise in HAI titers during the 2024–2025 season (Figs 2D, 3L). Participants aged 35–49 years had modest D0 HAI titers, but these titers increased following vaccination at D28 during the 2024–2025 season (Figs 2D and 3L). Participants aged 50–64 year had similar rise in HAI titers compared to younger participants against B/Victoria component in each season (Figs 2D, 3D, 3H and 3L). Finally, participants ≥65 years of age had the lowest D0 HAI titers against B/Yamagata compared to other age groups. However, these participants had the highest D28 post-vaccination HAI titers against B/ Victoria during 2022–2023–2023–2024 seasons (Figs 2D, 3D and 3H).

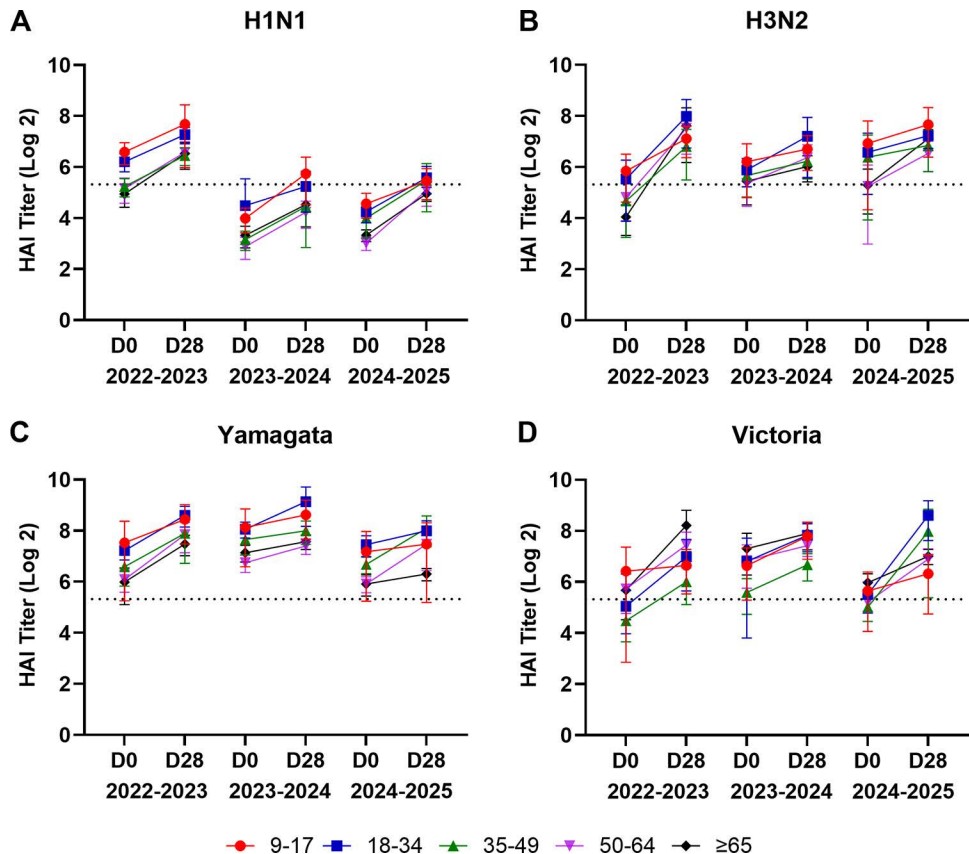

**Fig 2. Average hemagglutination inhibition (HAI) titers across all three cohorts in all participants.** Mean log$_2$-transformed HAI titers from serum samples collected at day 0 (D0, pre-vaccination) and day 28 (D28, post-vaccination) were measured against each seasonal vaccine component for cohorts UGA7 through UGA9. Overall seasonal cohorts are UGA7 n = 378; UGA8 n = 460; UGA9 n = 490 and participants are counted once per season. Data are presented for five age groups: 9-17 years, 18–34 years, 35–49 years, 50–64 years, and ≥65 years. Within each season, D0 and D28 values are connected within each age group to visualize within-season change; points denote means and error bars indicate 95% confidence intervals. Average log$_2$-transformed HAI titers are shown for (A) H1N1 vaccine component, (B) H3N2 component, (C) B/Yamagata lineage component, and (D) B/Victoria lineage component. The dotted horizontal line represents the seroprotection threshold (HAI titer 1:40, 5.32 log$_2$). Notably, the B/Yamagata lineage was not included in the UGA9 vaccine formulation; titers shown for B/Yamagata in 2024–2025 therefore reflect pre-existing immunity/back-boosting rather than a response to a vaccine component.

### Seroconversion and seroprotective antibody titers in sera collected from participants in all three seasons

The immune response following vaccination was assessed by analyzing serum hemagglutination inhibition (HAI) activity for rates of seroconversion (a four-fold rise in titer from Day 0) (Fig 4A), and the proportion of participants below the seroprotection threshold (HAI titer <1:40) (Fig 4B) or at/above seroprotection threshold (HAI titer ≥1:40) (Fig 4C). Between ~13%−25% of the vaccinated participants seroconverted following vaccination, over the three seasons against H1N1 (Fig 4A). Participants that entered each season with an HAI titer less than 1:40 HAI titer to each component were considered as non-seroprotective for that vaccine component. Approximately 31% of participants that were below the seroprotection threshold at D0 became seroprotective (≥1:40 HAI titer) against the H1N1component at D28 following vaccination (Fig 4B). Over the three seasons, among the Influenza A responses, participants had an average seroconversion rate of ~20% against the H1N1 component of the vaccine vs ~35% against H3N2 (Fig 4A). Among the influenza B vaccines, participants had the lowest average seroconversion rate (28%, 2024–2025 season excluded) against the B/

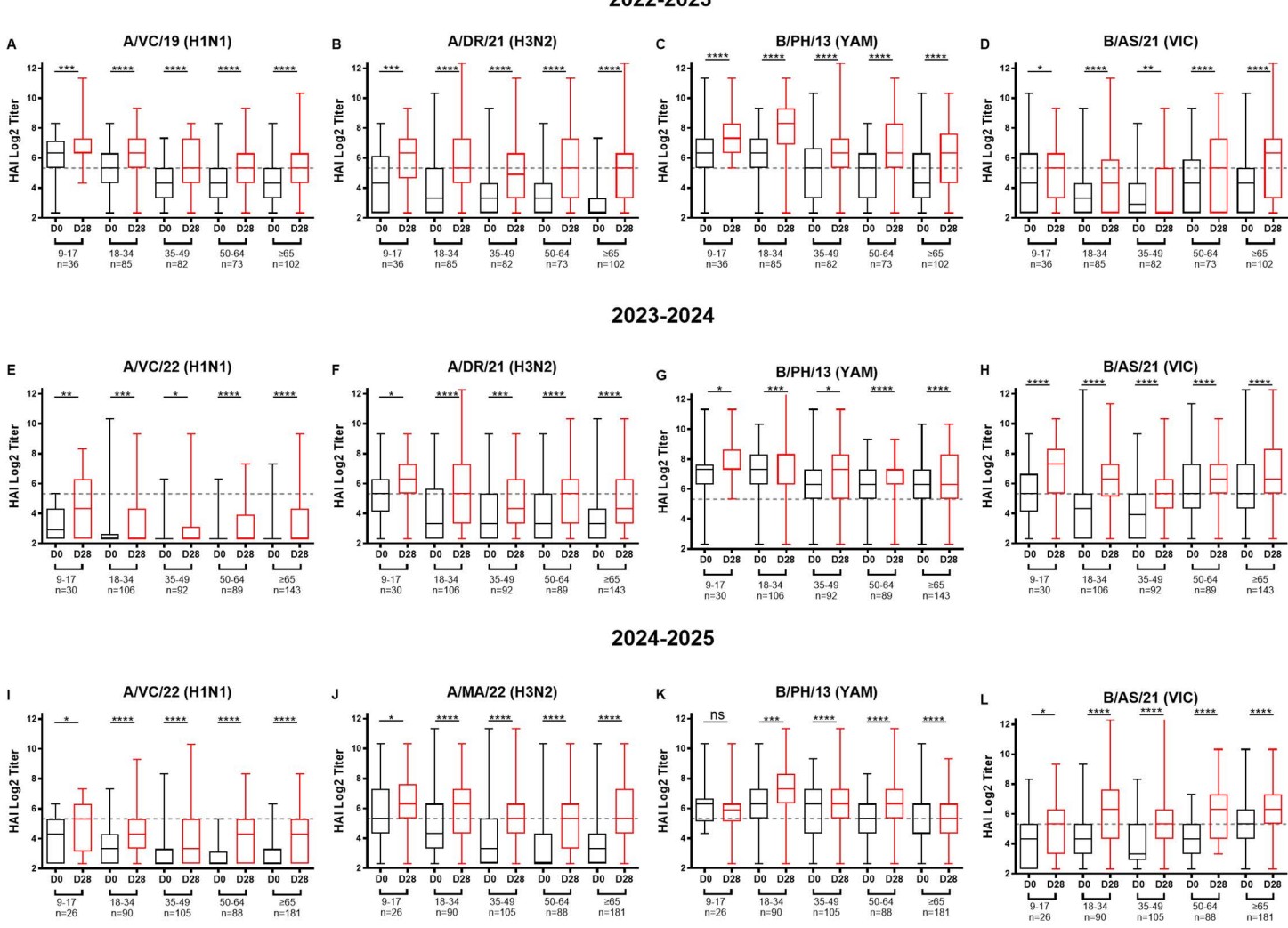

**Fig 3. Serum HAI responses to influenza vaccination across three consecutive seasons in overall study participants.** Hemagglutination inhibition (HAI) titers against the four vaccine components for each season are displayed as box-and-whisker plots across age groups at baseline (D0) and 28 days post-vaccination (D28). The box covers 50% of all values, with the lower quartile 1 (Q1) and upper (Q3) quartiles shown as the box limits, and the median indicated by a horizontal line. Comparisons between pre- and post-vaccination titers were assessed using the paired Wilcoxon signed-rank test (ns = not significant; *p < 0.05; **p < 0.01; ***p < 0.001; ****p < 0.0001). Data shown for overall seasonal study participants (2022–2023 n = 378; 2023–2024 n = 460; 2024–2025 n = 490). The number of individuals per age group is provided on below the x-axis in each panel. Results are shown for (A-D) 2022-2023, (E-H) 2023-2024, (I-L) 2024-2025 against vaccine components (A, E, I) H1N1, (B, F, J) H3N2, (C, G, K) B/Yamagata, and (D, H, L) B/Victoria. The dotted horizontal line represents the seroprotection threshold (HAI titer 1:40; 5.32 log$_2$). Notably, the B/Yamagata lineage was not included in the 2024-2025 vaccine formulation. However, HAI responses for B/Yamagata were shown for consistency across all cohorts; B/Yamagata titers in (K) are shown for continuity and reflect pre-existing immunity/back-boosting rather than responses to a vaccine component. Overall seasonal cohort sizes were: UGA7 n = 378; UGA8 n = 460; UGA9 n = 490.

Yamagata component compared to ~33% against the B/Victoria components across the three seasons (Fig 4A). Next, participants who were seroprotective at D0 and remained seroprotective following vaccination were examined (Fig 4C). Over the 3 seasons, on average ~10–25% of participants who were seroprotective at D0 seroconverted as determined by a 4-fold increase in HAI titers at D28 following vaccination against IBV (Fig 4C). Approximately 53–84% of the participants who were seroprotective at D0 were still seroprotective at D28 following vaccination. Some participants who were

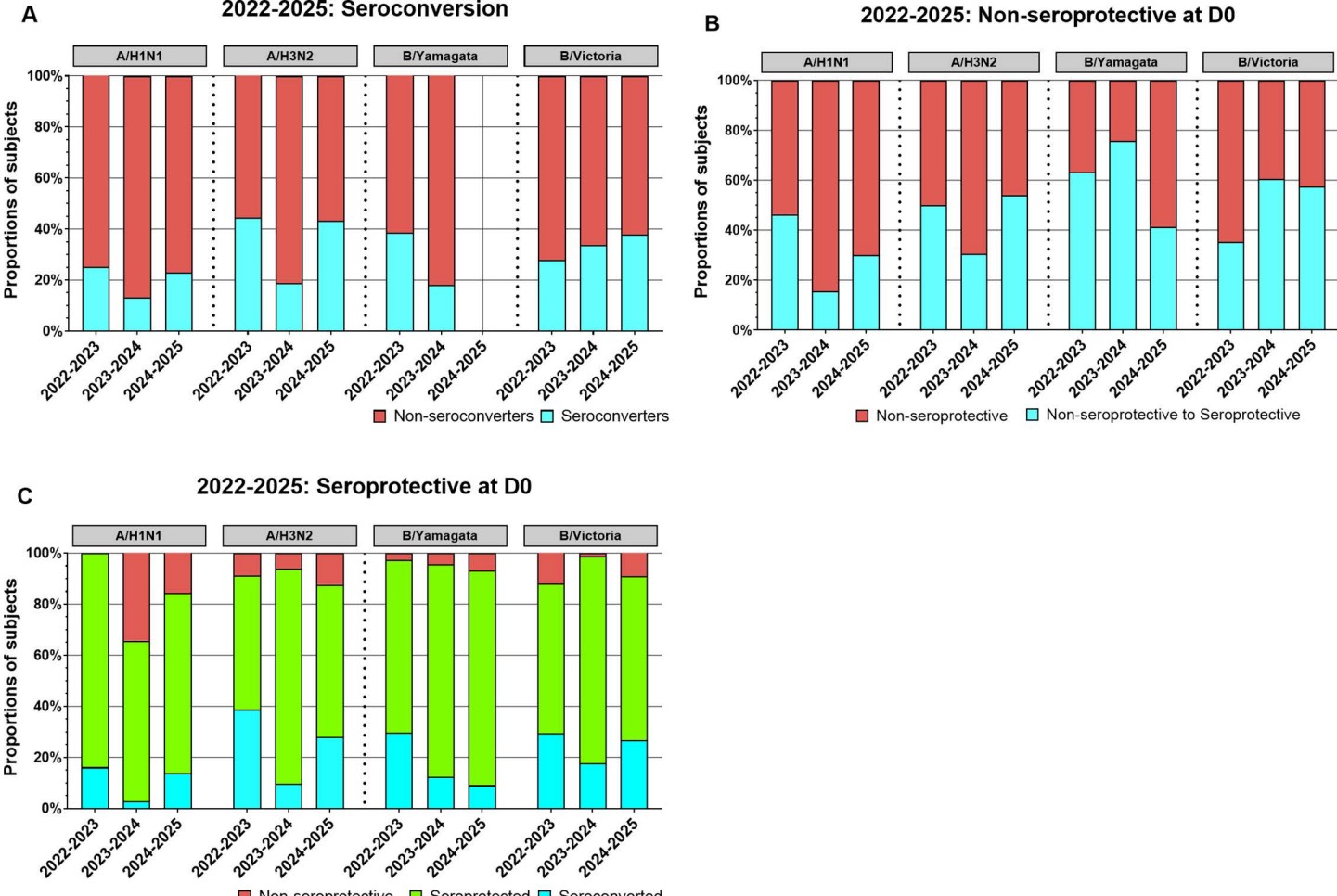

**Fig 4. Seroconversion and seroprotection profiles in UGA7–UGA9 overall study cohorts.** Stacked bar plots illustrate antibody responses to A/H1N1, A/H3N2, B/Yamagata, and B/Victoria for each season (2022–2023, 2023–2024, and 2024–2025), with bars showing the proportion of participants in each response category. Data shown for overall seasonal study participants (2022-2023 n = 378; 2023-2024 n = 460; 2024-2025 n = 490). (A) Individuals were categorized as seroconverters if they demonstrated a ≥ 4-fold increase in HAI titer from D0 to D28 and had a D28 titer ≥1:40 (blue), non-seroconverters are indicated in red. (B) Among participants who were non-seroprotective at baseline (D0 titer <1:40), bars show those who remained non-seroprotective at D28 (red) versus those who became seroprotective at D28 (D28 titer ≥1:40; blue). (C) Among participants with baseline seroprotective (titer ≥1:40 at D0) are separated into three groups: those who achieved seroconversion with a ≥ 4-fold rise at D28 (blue), those who maintained seroprotective titers without seroconversion (green), and those whose titers dropped below the seroprotective threshold at D28 (red). Notably, the B/Yamagata lineage was not included in the 2024-2025 vaccine formulation; therefore, seroconversion for this component could not be calculated in that cohort (A), as has also been indicated in other figures (B and C) for consistency (2022-2023, n = 378; 2023-2024, n = 460; 2024-2025, n = 490).

seroprotective at D0 were non-seroprotective following vaccination at D28. Fewer than 12% of the participants over the three seasons were listed in this category against the H3N2 and two influenza B components (Fig 4C). However, during the 2023–2024 season, ~34% of the participants that were seroprotective had a reduction in HAI titers after vaccination and were non-seroprotective against the H1N1 component at D28, whereas no participants were non-seroprotective against the H1N1 component during the 2022–2023 season (Fig 4C).

Seroconversion rates against the H1N1 component generally increased and varied by age, with ~30% of participants ≥65 years old seroconverting following vaccination during 2022 and 2023 (Fig 5A). During the 2024–2025 season,

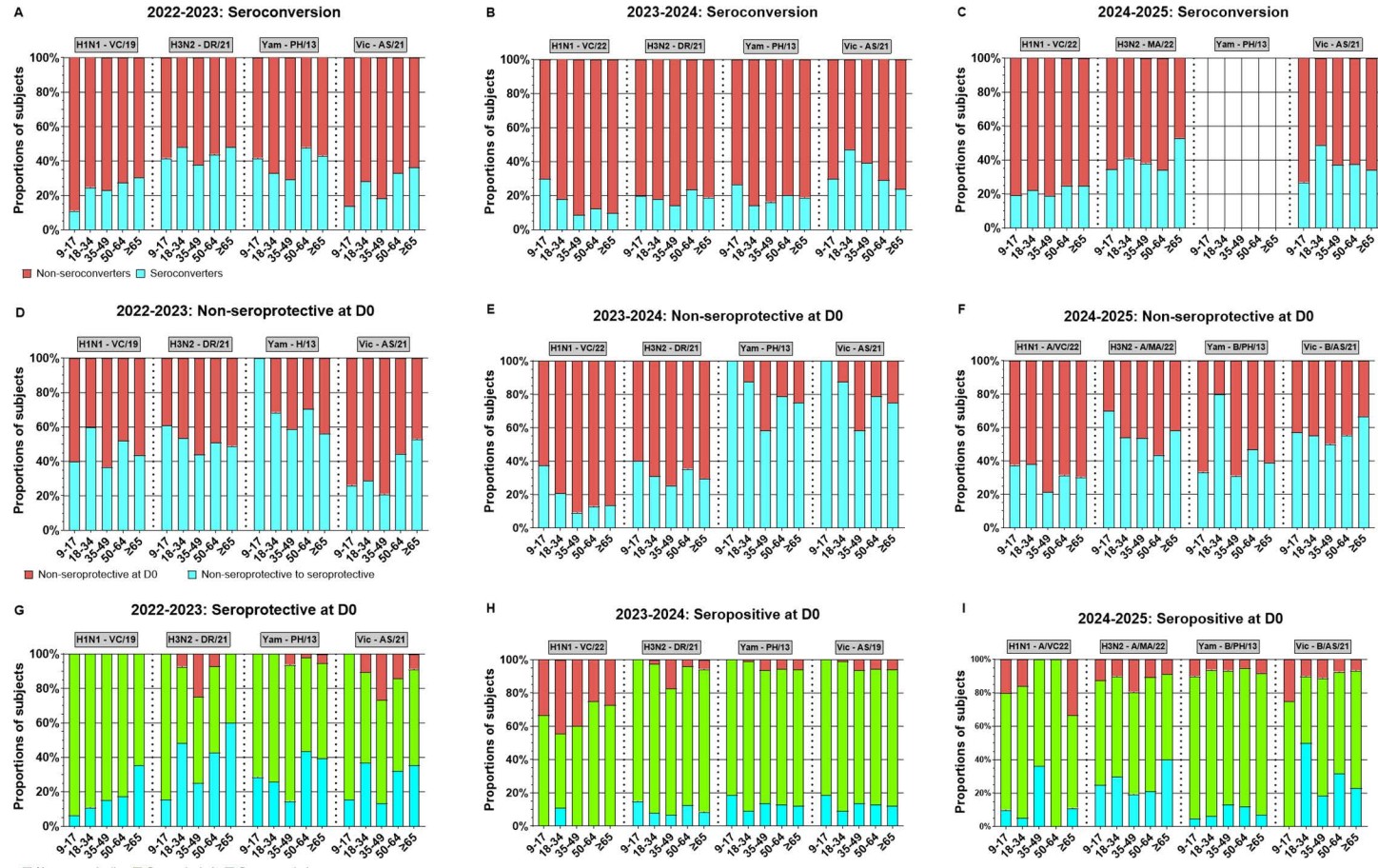

**Fig 5. Age-stratified seroconversion and seroprotection outcomes from sera collected overall study participants in each of the three seasons.** Stacked bar graphs illustrate hemagglutination inhibition (HAI) responses to H1N1, H3N2, B/Yamagata, and B/Victoria vaccine strains across three consecutive influenza seasons. Overall seasonal cohorts are 2022-2023 n = 378; 2023-2024 n = 460; 2024-2025 n = 490. Data are stratified by age groups within each cohort, as shown along the x-axis. (A–C) Seroconversion rates: participants achieving a ≥ 4-fold increase in HAI titer with a final value ≥1:40 at D28 are categorized as seroconverters (blue), whereas those not meeting this definition are shown as non-seroconverters (red). Notably, because the 2024–2025 vaccine formulation did not include a B/Yamagata component, seroconversion for this lineage could not be evaluated in panel C. (D–F) Outcomes for participants who were non-seroprotective at baseline (titer <1:40 at D0): individuals who remained non-seroprotective at D28 are indicated in red, while those who became seroprotective at D28 (≥1:40) represented in blue. (G–I) Responses among baseline seroprotective participants (titer ≥1:40 at D0): Bars show the proportion who seroconverted with a ≥ 4-fold rise (blue), those who retained seroprotective titers without seroconversion (green), and those whose titers declined below the seroprotective threshold by D28 (red). Note, the B/Yamagata lineage was not included in the 2024-2025 vaccine formulation; therefore, seroconversion for this component could not be calculated in that cohort (C), as has also been indicated in other figures (F and I) for consistency.

participants 50–64 and ≥65 years old had the highest seroconversion rates (~ 25%) and participants 35–49 years had the lowest (~19%) seroconversion rates (Fig 5A-5C). Seroconversion rates in 2022–2023 ranged ~38–48%, in 2023–2024 ranged from ~14–24%, and in 2024–2025, rates ranged between 34–53% against H3N2 (Fig 5A-5C). The youngest (9–17 years old) participants often had the largest portion of participants that moved from a non-seroprotective state at D0 to a seroprotective state at D28 (Fig 5D-5F). Similarly, among all participants, ~45% of those with non-seroprotective HAI titers against the H3N2 component on D0 were seroprotective at D28 across the three seasons (Fig 5D-5F) with younger participants having the highest rate of non-seroprotective HAI conversion to seroprotective HAI titers at D28 (Fig 5D-5F). For the influenza B viruses, over the 3 seasons ~60% of participants had non-seroprotective HAI titers against the B/

Yamagata component were converted to seroprotective at D28 (Fig 4B). Finally, over the 3 seasons ~51% of participants had non-seroprotective HAI titers against the B/Victoria component were seroprotective at D28 (Fig 4B). In general, younger participants had higher non-seroprotective to seroprotective rates than older participants for H3N2 across all seasons and against B/Yamagata in 2022–2023 and 2023–2024 seasons. In contrast, H1N1 conversion rates varied by season without consistent age-related trends, while B/Victoria conversion was often higher in older participants (≥65 years) than in younger participants (≤34 years) except during 2023–2024 season (Fig 5 and S6 Table).

### Impact of prior vaccination history on seroprotection and seroconversion to seasonal influenza vaccine components

Participants during the 2024–2025 season were stratified according to whether they had received influenza vaccination in the previous two seasons (2022–2023 [UGA7] and 2023–2024 [UGA8]) (Table 4). In the UGA9 cohort, a total of 201 participants had been vaccinated, while 114 participants had not been vaccinated in the previous two years. The D0 HAI serostatus and vaccine-induced HAI titers at D28 were compared across all four vaccine components (Table 4). In general, participants who had not been vaccinated in the prior two seasons had higher rates of seroconversion except for H3N2 and higher non-seroprotective to seroprotective transition rates following vaccination to multiple vaccine components. Among participants who were non-seroprotective at D0, those not vaccinated in the prior two seasons also had higher seroconversion rates than those vaccinated in the prior two seasons. Approximately 90% of participants were non-seroprotective to the H1N1 component at D0, regardless of whether they were vaccinated in the prior two seasons or not. Following vaccination, ~30% of participants not vaccinated in the prior two seasons seroconverted against H1N1 compared to ~24% of participants that were vaccinated in the prior two seasons. Among those participants that were non-seroprotective to the H1N1 component at D0, ~63% of these participants who were not vaccinated in the prior two seasons remained non-seroprotective while ~37% of these participants were seroprotective at D28. In contrast, among participants vaccinated in the prior two seasons, ~69% remained non-seroprotective against the H1N1 component at D0 and ~31% became seroprotective on D28 (Table 4). Among the participants that were seroprotective on D0, ~8% of participants who were not vaccinated during the prior two seasons seroconverted against H1N1, whereas ~24% of

**Table 4. Effects of prior two-season influenza vaccination history with baseline serostatus (D0) and post-vaccination response (D28) during the 2024–2025 season (UGA9).**

| Virus Subtype | Group | n-value | Total NSP at D0 | NSP at D0 & D28 | NSP at D0, SP at D28 | NSP at D0, SC at D28 | Total SP at D0 | SP at D0, NSP at D28 | SP at D0 & D28 | SP at D0, SC at D28 | SC |
|---|---|---|---|---|---|---|---|---|---|---|---|
| H1N1 | Vax 2 + Years | 201 | 89.6% | 68.9% | 31.1% | 23.9% | 10.4% | 14.3% | 61.9% | 23.8% | 23.9% |
| | No Vax 2 Years[a] | 114 | 88.6% | 63.4% | 36.6% | 32.7% | 11.4% | 30.8% | 61.5% | 7.7% | 29.8% |
| H3N2 | Vax 2 + Years | 201 | 78.6% | 48.7% | 51.3% | 47.5% | 21.4% | 7.0% | 46.5% | 46.5% | 47.3% |
| | No Vax 2 Years | 114 | 64.0% | 37.0% | 63.0% | 58.9% | 36.0% | 14.6% | 63.4% | 22.0% | 45.6% |
| YAM | Vax 2 + Years | 201 | 37.3% | 65.3% | 34.7% | 16.0% | 62.7% | 7.1% | 86.5% | 6.3% | 10.0%[b] |
| | No Vax 2 Years | 114 | 43.9% | 40.0% | 60.0% | 38.0% | 56.1% | 7.8% | 73.4% | 18.8% | 27.2%[b] |
| VIC | Vax 2 + Years | 201 | 64.7% | 43.1% | 56.9% | 43.8% | 35.3% | 12.7% | 66.2% | 21.1% | 35.8% |
| | No Vax 2 Years | 114 | 49.1% | 28.6% | 71.4% | 60.7% | 50.9% | 5.2% | 62.1% | 32.8% | 46.5% |

Table shows UGA9 subset (n = 315) with available the prior two-season vaccination history (no influenza vaccination received in the two preceding seasons n = 114; two or more influenza vaccination received in the two preceding seasons n = 201). Outcomes are shown for each vaccine component tested (H1N1, H3N2, VIC, YAM).

YAM, Yamagata lineage; VIC, Victoria lineage; SP, seroprotective (HAI ≥ 1:40); NSP, non-seroprotective (HAI < 1:40); SC, seroconversion (≥4-fold rise from D0 with D28 ≥ 1:40); D0, day 0; D28, day 28. Percentages are calculated within each row unless otherwise indicated.

a No Vax 2 Years = no influenza vaccination received in the two preceding seasons (2022–2023 and 2023–2024).

bSeroconversion against B/Yamagata in the 2024–2025 season was assessed using HAI responses to the historical strain B/Phuket/3073/2013 (B/PH/13) despite its exclusion from the 2024–2025 vaccine formulation.

participants vaccinated in the prior two seasons had seroconversion (Table 4). Similar results were observed against the H3N2 component of the vaccine. Approximately 64% of participants not vaccinated in the prior two seasons and ~79% of those participants vaccinated in the prior two seasons were non-seroprotective at baseline against H3N2. Among participants who were seroprotective at D0, ~47% of participants vaccinated during the prior two seasons seroconverted against H3N2, compared with 22% of those participants not vaccinated in the prior two seasons. However, overall seroconversion rates were similar (~46%) in both groups at D28 (Table 4).

Distinct differences emerged between participants that were vaccinated and not vaccinated during two prior seasons against the influenza B lineages (B/Yamagata and B/Victoria). Approximately 37% of participants who were vaccinated in the prior two seasons and ~44% of participants that were not vaccinated during the prior two seasons were non-seroprotected at D0 to the B/Yamagata virus. Following vaccination, ~10% of participants vaccinated in the prior two seasons seroconverted compared with ~27% of participants who were not vaccinated in the prior two seasons. Among participants who were seropositive at D0, ~87% of vaccinated participants and ~73% of non-vaccinated participants remained seropositive following vaccination. Approximately 65% of participants who were vaccinated in the prior two seasons were non-seroprotected at D0 against the B/Victoria component in the vaccine compared to ~49% of participants who were not vaccinated in the prior two seasons. Following vaccination, ~44% of these participants who were vaccinated in the prior two seasons seroconverted compared with ~61% of participants who were not vaccinated in the prior two seasons. Among participants who were seropositive at D0 against the B/Victoria component, ~21% of those vaccinated during the prior two seasons seroconverted, whereas ~33% of those not vaccinated in the prior two seasons seroconverted. The vast majority of these participants maintained a seropositive status at D28 (Table 4).

Overall, participants who were not vaccinated in the prior two seasons showed higher seroconversion rates and higher non-seroprotective to seroprotective transition rates following vaccination against H1N1 and the influenza B vaccine components. However, participants vaccinated in the prior two seasons had slightly higher baseline seroprotection rates against B/Yamagata, whereas participants who skipped vaccination had higher seroconversion responses against H1N1 and the influenza B components, whereas H3N2 seroconversion rates were similar between groups. These findings indicate that prior vaccination history influenced both baseline immunity and vaccine responsiveness with larger post-vaccination responses observed against H1N1 and influenza B viruses among participants who had not been vaccinated in the prior two seasons (Table 4).

**Longitudinal seroprotection in the sera collected from participants in all three seasons.** A longitudinal analysis was conducted on sera collected from participants from all three seasons that was evaluated for HAI seroprotection at six time points over a three-year period. Seroprotection was assigned a score of "1" and non-seroprotection a score of "0" for each time point, resulting in a cumulative score ranging from 0 to 6 for each participant. A score of 6 represented participants who maintained seroprotective HAI titers great than 1:40 at all six time points (D0 and D28 for in each of the three seasons) (Figs 6-7 and S7 Table).

**Influenza A (IAV) seroprotection: H1N1 and H3N2 components.** Participants between the ages of 9–17 years old had several number of participants with a score of 6 against H1N1 and H3N2 strains with 50% of the participants having a score of 4 or higher against H1N1 viruses and 64% against the H3N2 viruses (Fig 6A-B). In contrast, few adults over the age of 18 years old had a score of 6 against H1N1 components however, there were low number of adults in all age groups that had a score of 6 (between 2–14%) against the H3N2 viruses. No participants under the age of 17 years were non-seroprotective (score of 0) at any of the 6 timepoints assessed against the H1N1 component (Fig 6A). However, 18–33% of adults per age group were non-seroprotective at all 6 time points with a score of 0 against the H1N1 components (Fig 6A). Similar results were observed against the H3N2 components, with no participants 9–17 years had a score of 0, while 14% had score of 1 (Fig 6B).

### Influenza B (IBV) seroprotection: B/Yamagata and B/Victoria components

For the influenza B viruses, the majority of the participants in younger age groups (9–17: 64.3%; 18–34 years: 64.3%) had a score of 6 against the B/Yamagata components, whereas participants over the age of 65 years had the lowest

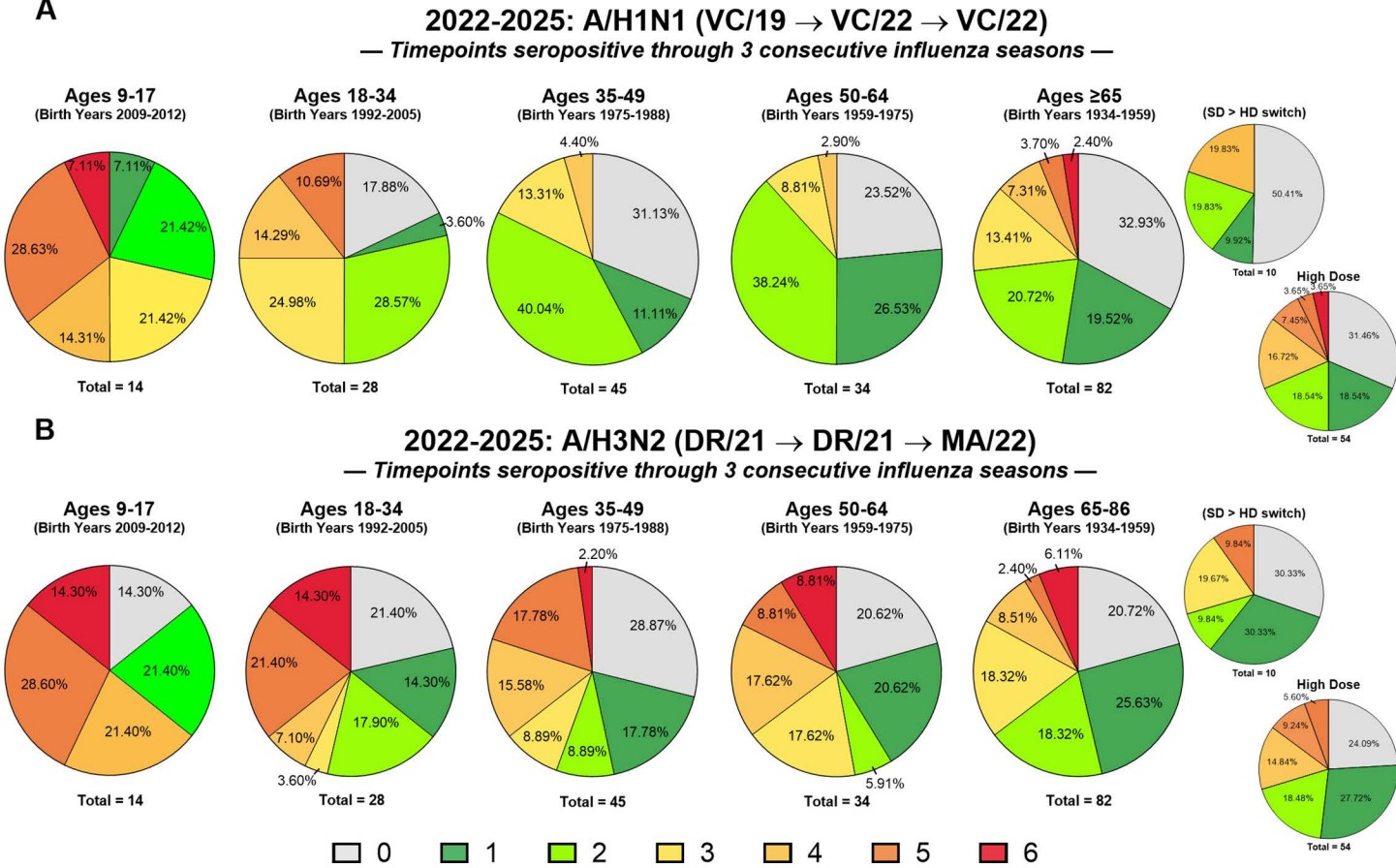

**Fig 6. Influenza A virus (IAV) seroprotective patterns across three consecutive influenza seasons.** A total of 203 longitudinal repeater participants vaccinated across the 2022–2023, 2023–2024, and 2024–2025 seasons were evaluated for baseline and post-vaccination antibody titers to (A) A/H1N1 (VC/19→VC/22→VC/22) and (B) A/H3N2 (DR/21→DR/21→MA/22), as indicated above each panel. Seroprotection was measured at six time points (D0 and D28 within each season). Each individual was assigned a cumulative score ranging from 0 to 6, reflecting the number of time points at which they achieved an HAI titer ≥1:40. A score of 0 indicated no seroprotective titers at any time point, while a score of 6 represented sustained seroprotection across all measurements. Pie charts display the proportion of participants in each age group at each cumulative seroprotection score (0–6); colors correspond to cumulative scores (0–6) as shown in the key. The total number of participants in each age group indicated below the corresponding pie chart. For participants aged 65–86 years, results are displayed both as an aggregate and subdivided by vaccine formulation: high-dose (HD), standard-dose (SD), or those who switched from SD to HD across the study period.

percentage (~36%) (Fig 7A). Few participants were non-seroprotective against the B/Yamagata component with a score of 0 at all 6 timepoints (Fig 7A). Compared to B/Yamagata, fewer participants had a score of 6 and more participants were non-seroprotective with a score of 0 against the B/Victoria component and the percentage of non-seroprotective participants decreased with age (Fig 7B).

**Comparison of HAI activity in collected sera from participants vaccinated over the three seasons.** Vaccinated participants had distinct differences in vaccine-induced antibody responses against each vaccine component depending on the type of vaccine administered (Fig 8 and S5 Table). Participants vaccinated with any of the six vaccines (Fluzone Standard Dose, Fluzone High-Dose, Fluad, Flublok, Flucelvax, and Flumist) had, on average, a statistical rise in HAI titers against the H1N1 vaccine component over the three seasons, except for participants vaccinated with Flumist (Fig 8A). Participants vaccinated with Flublok had the highest seroprotective

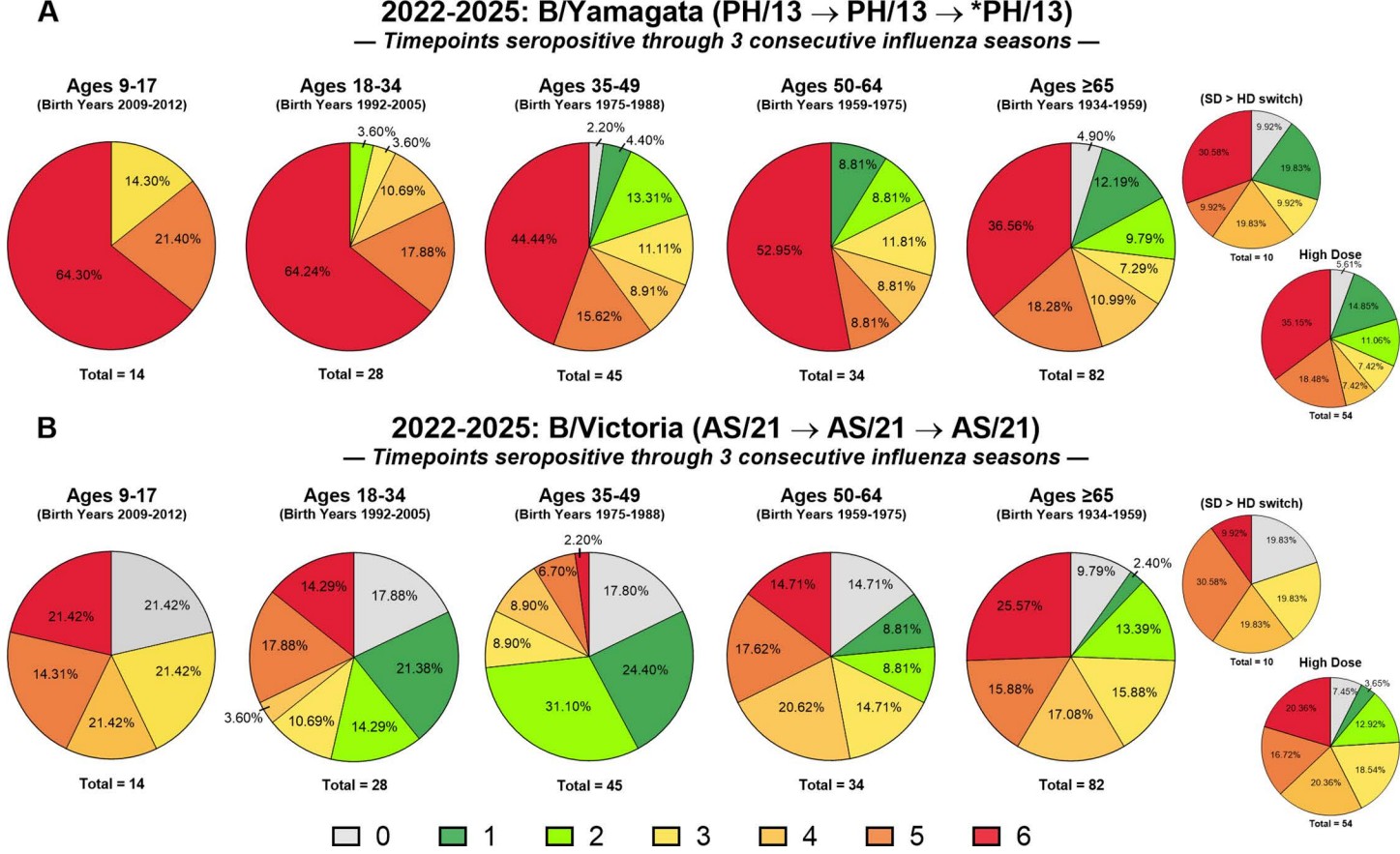

**Fig 7. Influenza B virus (IBV) seroprotection patterns across three consecutive influenza seasons.** A total of 203 longitudinal repeater participants vaccinated across the 2022–2023, 2023–2024, and 2024–2025 seasons were evaluated for antibody responses to (A) B/Yamagata and (B) B/Victoria at both D0 0 and D28. Seroprotection was assessed at six time points (D0 and D28 in each season), and individuals were assigned a cumulative score from 0 to 6, corresponding to the number of time points with a HAI titer ≥1:40. A score of 0 indicated no seroprotective titers across all time points, whereas a score of 6 reflected persistent seroprotection throughout the study period. Pie charts display the proportion of participants in each age group at each cumulative seroprotection score (0–6); colors correspond to cumulative scores (0–6) as shown in the key. Distributions are presented by age group, with the total number of participants in each age group indicated below the corresponding pie chart. Participants aged 65–86 years are displayed both as a combined group and subdivided by vaccine formulation: high-dose (HD), standard-dose (SD), or those who switched from SD to HD during the study. Note: the 2024–2025 vaccine did not include a B/Yamagata component; results are shown against the WHO-recommended strain for continuity of analysis.

HAI titer at D28 following vaccination compared to participants vaccinated with any of the other 5 vaccines. These same participants showed substantially higher HAI titers against H3N2 compared to H1N1 (Fig 8B). Although B/ Yamagata was not included in the 2024–2025 vaccine formulation, HAI titers were still measured to align with the other seasons. On average, baseline D0 HAI titers were higher against the B/Yamagata component in adult participants each season compared to elderly participants (Fig 8C). While B/Yamagata excluded from the vaccine, HAI titers against B/Yamagata component increased from D0 to D28 during the 2024–2025 and participants vaccinated with Fluzone SD, Flucelvax, and Fluzone HD had a statistically significant overall (2022–2023– 2024–2025) rise in HAI titers at D28 (Fig 8C). Similar vaccine type results were observed against the B/Victoria component (Fig 8D).

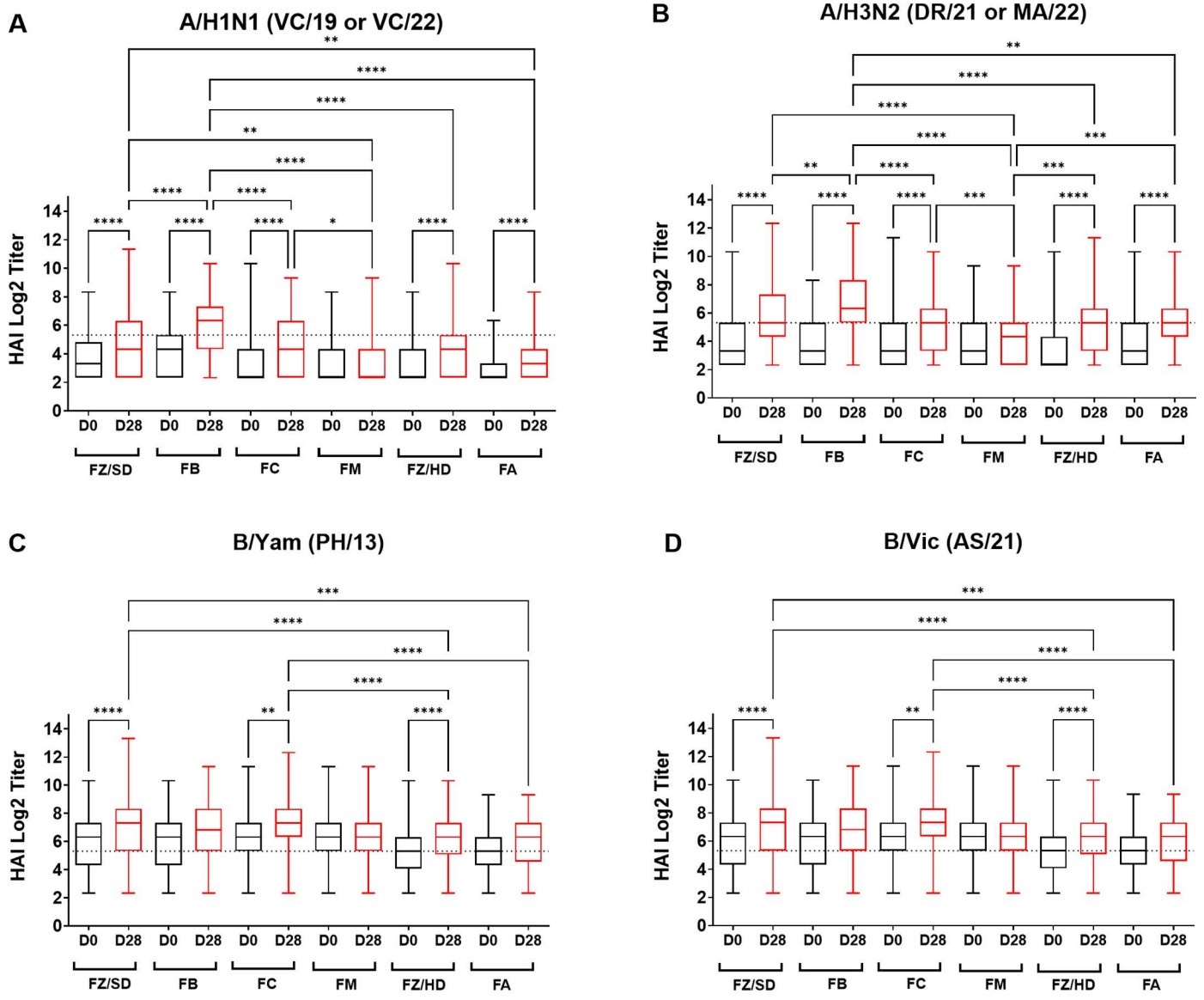

**Fig 8. HAI responses by vaccine type across three seasons (2022-2023 to 2024-2025).** Panels show HAI immune responses against (A) H1N1 (VC/19 during 2022–2023 and VC/22 during 2023–2025), (B) H3N2 (DR/21 during 2022–2024 and MA/22 during 2024–2025), (C) B/Yamagata (B/PH/13 during 2022–2025), and (D) B/Victoria (AS/21 during 2022–2025). Boxplots show $\log_2$ HAI titers at baseline (D0) and post-vaccination (D28) for each vaccine: FZ/SD (Fluzone Standard Dose), FB (Flublok), FC (Flucelvax), FM (Flumist), FZ/HD (Fluzone High-Dose), FA (Fluad). Horizontal dotted line indicates the seroprotective threshold of 1:40 HAI titer ($\log_2 \sim 5.32$). Comparisons between D0 and D28 within vaccine groups and between vaccine types at D28 were assessed by one-way ANOVA with Games-Howell's multiple comparisons test after HAI titers were $\log_2$-transformed [$y = \log2(y)$]: $p < 0.05$ (*), $< 0.01$ (**), $< 0.001$ (***), $< 0.0001$ (****). Sample sizes: FZ/SD n = 453, FB n = 102, FC n = 246, FM n = 129, FZ/HD n = 270, FA n = 128. Vaccine components were tested against A/H1N1 (Vic/19 during 2022-2023 and VC/22 during 2023-2025), H3N2 (DR/21 during 2022-2024 and MA/22 during 2024-2025), B/Yamagata (B/PH/13 during 2022-2025), B/Victoria (AS/21 during 2022-2025).

### Age and vaccine specific variability in HAI responses

Responses to each type of vaccine were stratified by each for seroprotective HAI titers against a panel of influenza viruses (Fig 9 and S8 Table). Across all three seasons, there was an overall reduction in vaccine-elicited HAI responses

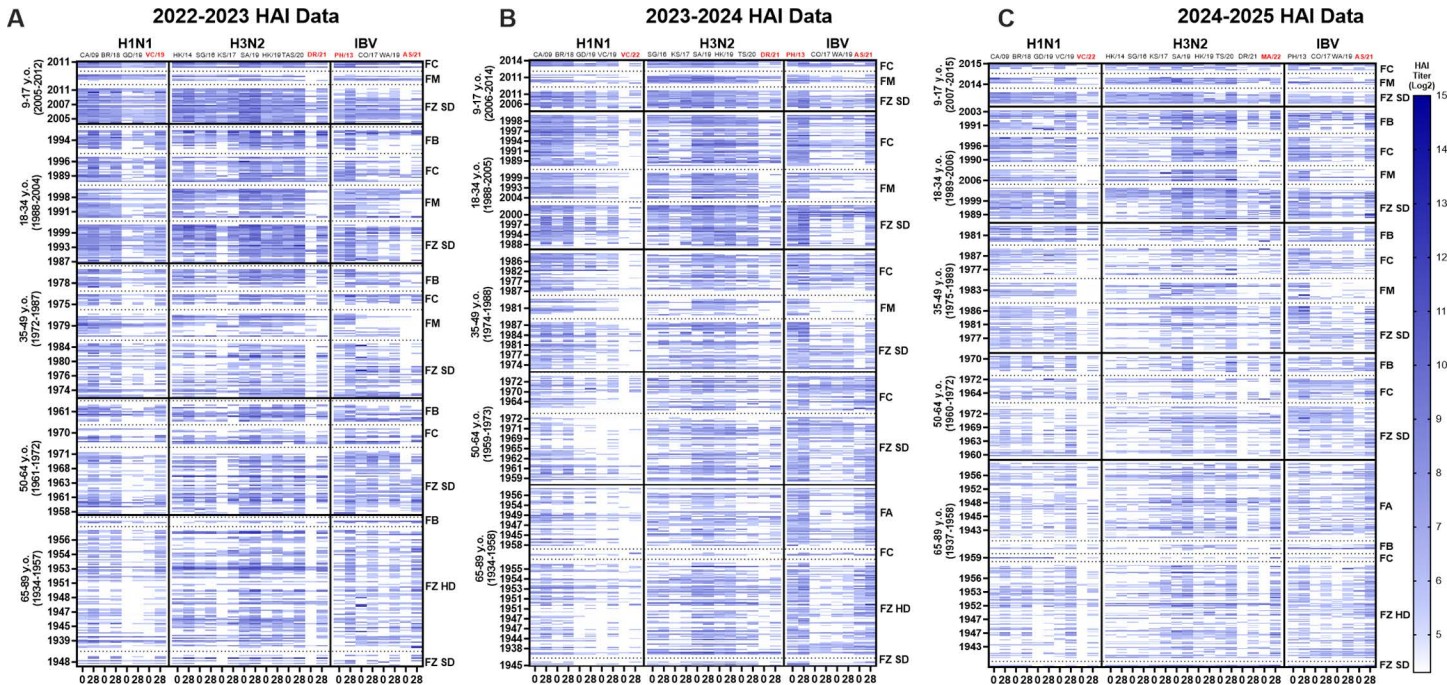

**Fig 9. Heat map analysis of hemagglutination inhibition (HAI) titers from sera collected from participants in all the three seasons.** Serum specimens collected at baseline (D0) and 28 days post-vaccination (D28) were evaluated against panels of historical influenza A and B viruses. Heat maps are shown for overall study participants; (A) the 2022–2023 season (n = 378), (B) the 2023–2024 season (n = 460), and (C) the 2024–2025 season (n = 490). Within each panel, participants are stratified by age group (9-17, 18–34, 35–49, 50–64, and ≥65 years) and separated by vaccine formulation (Fluzone SD, Fluzone HD, Flublok, Flucelvax, Fluad, and Flumist), with individual birth years indicated along the y-axis. Viral antigens are ordered chronologically along the x-axis, from the earliest to the most recent isolates and are grouped by subtype/lineage (H1N1, H3N2, and IBV) as indicated above each panel. Within each subtype/lineage block, paired heat maps display D0 (left) and D28 (right) responses. Vaccine components are highlighted in red text at the top of the heat maps. The B/Yamagata lineage was not included in the 2024-2025 vaccine formulation (panel C). HAI titers are expressed as $\log_2$ values, with HAI titers ≥1:40 represented by a graded blue color scale (lighter shades = lower titers; darker shades = higher titers) and titers <1:40 (non-seroprotective) displayed in white.

were in the 2023–2024 season (Fig 9B) compared to 2022–2023 season (Fig 9A), with an even further decline during the 2024–2025 season (Fig 9C). During the 2022–2023 season, Fluzone SD elicited broad activity across both vaccine components and historical strains (Fig 9A), with Fluzone HD inducing robust increases in ≥65 years of age, while Fluzone SD was less effective. Flublok generated strong responses in 18–34 years of age, especially against H1N1 and H3N2, though cross-reactivity diminished with age (Fig 9A). Flucelvax provided modest increases, most evident against H1N1 and some historical H3N2 viruses, whereas Flumist produced minimal responses across groups (Fig 9A). During the 2023–2024 season, Fluzone SD again produced broad activity against both vaccine and historical strains and Fluzone HD induced strong serum HAI activity in ≥65 years of age (Fig 9B). Fluad, administered to elderly participants, also elicited robust and broad cross-reactive responses across H1N1, H3N2, and both B lineages, minimizing seroprotection threshold (HAI < 1:40) (Fig 9B). Flucelvax induced moderate, age-dependent activity, with responses most consistent in 18–64 y.o., while Flumist was less effective especially against vaccine components (Fig 9B). During the 2024–2025 season, overall responses were lower, but Fluzone HD and Fluad still produced strong boosting in ≥65 years of age with broad HAI activity against historical strains (Fig 9C). Notably, despite exclusion of B/Yamagata from the 2024–2025 vaccine formulation, there were increases in cross-reactive HAI titers in sera collected from participants of different ages, particularly from older participants vaccinated with Fluzone HD and Fluad (Fig 9C). Flublok maintained strong HAI titers in participants between

the ages of 18–34 and 35–49 years, primarily against the H1N1 and H3N2 vaccine components, while Flucelvax elicited modest, strain-specific HAI activity, and Flumist again failed to elicit seroprotective HAI titers (Fig 9C).

## Discussion

In this study, HAI antibody titers were investigated over three seasons (2022–2023–2024–2025) in participants vaccinated with different influenza virus vaccines (Tables 1 and 2). A total of 1,328 participants aged 9–89 years were enrolled with informed consent and a "repeater cohort" of 203 participants were enrolled during all three seasons (Table 3). To improve interpretability of this large observational dataset, we address four primary questions: (i) how baseline (D0) and post-vaccination (D28) HAI responses vary by age group across consecutive influenza seasons; (ii) within indicated populations, how licensed vaccine formulations compare in measured HAI responses; (iii) how baseline titers influence observed post-vaccination fold-rise and seroconversion metrics; and (iv) how prior-season vaccination history and repeat vaccination patterns (including the longitudinal repeater cohort) are associated with baseline titers and post-vaccination responses. Consistent with previous reports, repeated prior vaccination history is associated with attenuated HAI boosting responses, but it also confers higher baseline immunity and better maintenance of seroprotection [12,13,16,17].

With respect to age-stratified immunogenicity across seasons, age was a major determinant of immunogenicity. Older adults (≥65 years of age) had lower baseline D0 HAI titers with smaller post-vaccination rises in HAI titers compared to younger participants [6]. Across age groups, D28 titers were generally followed by lower titers at the subsequent season's pre-vaccination baseline (D0), consistent with between-season decline; this decline was more pronounced in older adults, in whom D0 titers at the subsequent season more often fell below the seroprotection threshold (HAI < 1:40) [18]. In our cohorts, increasing mean age of enrolled participants across the three seasons (from 47.1 to 52.4 years; Table 1) represents a demographic confounder that may contribute to apparent declines in aggregate HAI responses in later seasons, independent of vaccine composition changes. However, age-stratified analyses within each season control for this confounding effect, and the longitudinal repeater cohort analysis tracks the same individuals over time, mitigating concerns about between-season age confounding.

Regarding comparisons of vaccine formulations within indicated populations, in adults and children (<65 years of age), recombinant HA vaccines (Flublok) induced stronger IAV HAI titers than other vaccine types, while IBV HAI responses were generally comparable to those elicited by egg-based vaccines (Fluzone SD). In contrast, the live-attenuated intranasal vaccine (FluMist) often generated lower serum HAI antibody titers for both IAV and IBV following vaccination, consistent with previous studies [19]. In older adults over the age of 65, the high-dose inactivated split vaccine (Fluzone HD) and MF59-adjuvanted vaccine (Fluad) elicited similar HAI titers following vaccination.

In this study, younger participants, especially participants between the ages of 9–17 years, had higher baseline D0 HAI titers and broader cross-reactivity to historical influenza strains compared to older adults (Figs 1 and 2). After vaccination, these younger individuals mounted robust antibody titers with HAI activity, not only to vaccine-matched strains, but also to historical viral variants. These antibodies against historical H1N1 influenza strains persisted at protective levels for up to 12 months. In contrast, many elderly participants (≥65 years old) had the lowest baseline HAI titers and a much lower breadth of reactivity prior to vaccination compared to younger participants. However, following vaccination, these elderly participants had seroprotective HAI titers at D28 (HAI ≥ 1:40) that were lower at the next season's baseline (D0), consistent with between-season decline (Figs 1 and 2). Elderly participants also had lower D0 HAI titers prior to vaccination each season compared to younger participants. Because fold-rise–based metrics depend strongly on baseline titers, lower D0 values in older adults contributed to higher apparent seroconversion frequencies (≥4-fold rise) despite lower absolute titers compared with younger age groups. Often, participants below the seroprotection threshold (HAI < 1:40) converted to seroprotective following vaccination, whereas younger participants that were already seroprotective had smaller fold increases in HAI titers. This pattern, also noted in earlier UGA cohorts in our previous study, indicates a consistent age-related HAI decline and waning trend [12]. This may be explained by immunosenescence in the elderly, a condition

characterized by diminished B-cell function, reduced generation of high-affinity antibodies, and impaired immune memory, all of which limit the magnitude and durability of vaccine-induced responses [20–23]. Moreover, long-standing immunologic memory in older adults may bias responses toward epitopes located in historical influenza strains, a phenomenon consistent with original antigenic sin [24].

To address how baseline titers influence fold-rise and seroconversion metrics, participants who had not received influenza vaccination in the previous two years had higher overall seroconversion rates (~30% vs ~24%) against H1N1, while H3N2 rates were comparable (~46% vs ~47%) compared to those vaccinated in consecutive seasons (Tables 4 and S6). However, when participants were stratified by their baseline (D0) serostatus, an opposite trend emerged among those who were below the seroprotection threshold (HAI < 1:40) before vaccination. Among participants who were seroprotective at baseline, those with consecutive annual vaccinations exhibited higher proportions achieving a ≥ 4-fold rise post-vaccination for influenza A viruses (H1N1: ~24% vs ~8%; H3N2: ~47% vs ~22%), whereas for influenza B/Victoria, a smaller proportion achieved such rises compared to those without prior vaccination (~21% vs ~33%) season (Tables 4 and S6). It was previously reported that participants vaccinated annually over multiple years had lower post-vaccination HAI titers against vaccine strains each season, whereas participants who skipped a season or two mounted more vigorous responses [12,13,15,25]. Healthcare workers during 2020–2021 influenza season, that had been vaccinated each of the prior five years, had the lowest post-vaccination titers against influenza A viruses and people not vaccinated or vaccinated a single time in prior influenza seasons had the highest HAI titers [15]. Participants with elevated pre-vaccination HAI titers at D0 had smaller increases in HAI titer following vaccination. This is termed a "ceiling effect" [26–28], whereby residual immunity from repeated exposures can dampen increases in the antibody titers following a new vaccine. Regarding prior-season vaccination history and repeat vaccination patterns (including "skipped years"), our data support descriptive associations between prior-season vaccination history and measured baseline titers and post-vaccination fold-rise. However, because this is an observational study, it cannot quantify the clinical tradeoff between vulnerability during an unvaccinated season and any potential boosting after a gap, and it should not be interpreted as evidence to change existing vaccination recommendations.

Alternatively, the lack of antibody responsiveness following vaccination may be due to the antigenic distance hypothesis [14,29,30]. When vaccines from consecutive years are antigenically similar, repeated exposure can focus the immune response on previously recognized epitopes rather than generating new antibodies to a diverse set of epitopes (Fig 1). This phenomenon limits responses to novel antigenic sites on contemporary strains [14,31]. As a consequence, negative interference associated with repeated vaccinations using similar vaccines may occur and antibodies can be less effective if circulating strains have drifted away from prior vaccine strains. Overall, annual vaccination may refocus immune memory on conserved or previously encountered epitopes ("original antigenic sin") [24,32]. This phenomenon occurs only when there is antigenic similarity between the priming and boosting strains, whereas no recall responses are observed when the boosting strain is antigenically distant from the priming strain [33].

In this study, updating influenza A vaccine strains significantly impacted the elicitation of immune responses between the new and old strains. There was a distinct HAI response patterns for influenza A (H1N1 and H3N2) and B viruses. VC/19 (clade 6B.1A.5a.2) was the H1N1 vaccine component used in the 2022–2023 influenza vaccines while VC/22 (clade 6B.1A.5a.2a.1) was used in the 2023–2024 and 2024–2025 influenza vaccines. Rates of seroconversion were highest during the 2022–2023 (~25%) and 2024–2025 (~23%) seasons, but lower during the 2023–2024 (~13%) season (Fig 5A-5C). The drop during the 2023–2024 season may be due to the amino acid differences between the H1 HA antigens used in the vaccine resulting in lower D0 baseline titers and low rates of seroconversion. H1N1 influenza viruses of the clade 6B.1A.5a.2 contain D260E and T277A substitutions in the HA head region, as well as P137S and K142R changes in the receptor-binding region [34,35]. These changes are associated with antigenic drift, reducing cross-reactivity to antibodies elicited by earlier clades [36]. Accordingly, participants vaccinated during the 2023–2024 season had lower D0 HAI titers against VC/22 than to VC/19, yet there were measurable serum HAI (≥1:40) titers following

vaccination in the following season. At D0, participants had HAI activity against historical H1N1 strains, CA/09, clade 1 and BR/18, clade 6B.1A.1, that were back-boosted following vaccination indicating durable memory to antigenically conserved epitopes on HA.

In this study, H3N2 components had more stable seroconversion patterns across seasons without an antigenic update in 2023–2024, as many participants were already seroprotected at D0 compared to the H1N1 components in each vaccine among those seroprotective at D0 (Fig 5G-5I). During the 2023–2024 influenza season, DR/21 (clade 3C.2a1b.2a.2) was included as the H3N2 vaccine component, whereas MA/22 (clade 3C.2a1b.2a.3a.1) was included as the H3N2 vaccine component during the 2024–2025 season. MA/22 HA has an "N" at position 122, whereas the DR/21 HA used in the previous season had the amino acid "D". This change resulted in a loss of putative glycosylation site in MA/22 in the HA at this position compared to DR/21. In addition, there is a K to E change at amino acid 276 that may reduce antibody binding of antibodies elicited against DR/21 HA (3C.2a1b.2a.2) viruses [37]. Younger participants showed large post-vaccination HAI titer increases, whereas the elderly showed only moderate increases. These vaccinated participants had increases in HAI titers against SA/19, HK/19, and TS/20, even though these antigens were not included in the vaccines during these three seasons. During 2024–2025 season, HAI activity against older strains, such as HK/14, SG/16, KS/17 was markedly lower than previous seasons. Overall, these findings indicate that accumulating head-epitope substitutions, especially at dominant antigenic sites, narrow HAI cross-reactivity and limit recall to antigenically distant strains and limit HAI back-boosting [38–40].

Participants vaccinated during the 2022–2023 influenza season had relatively high HAI titers and seroconversion against the B/PH/13 (Yamagata) virus compared to HAI titers against this virus during the 2023–2024 season (Fig 5A and 5B). Although the vaccines no longer included a B/Yamagata component, these antibodies with HAI activity persisted into the 2024–2025 season. However, 25% of participants who were non-seroprotective at D0 against the Yamagata during the 2024–2025 season converted to seroprotective status at D28—substantially lower than the 50% and 52% conversion rates observed in 2022–2023 and 2023–2024, respectively when B/Yamagata was included in the vaccine (Fig 5C, 5F and 5I) [37]. This finding is consistent with reports that B/Yamagata viruses have not been isolated after 2020, yet measurable immunity to B/Yamagata can persist, likely reflecting prior exposures and/or cross-reactive boosting from B/Victoria infections or vaccinations [41]. This is most likely due to immunity elicited by influenza vaccines or viral infections from previous seasons. Importantly, while HA-directed HAI responses are typically lineage-specific, reports describing the disappearance of B/Yamagata circulation since 2020 yet persistence of B/Yamagata-reactive antibodies suggest that cross-reactive recall (including neuraminidase-mediated immunity) following B/Victoria infection or vaccination may contribute to maintaining measurable B/Yamagata-directed immunity after removal of the B/Yamagata vaccine component [41]. In experimental models, cross-lineage protection after B/Victoria vaccination has been reported to be stronger against B/Yamagata challenge than the reverse, supporting the plausibility that repeated exposure to B/Victoria vaccine antigens could contribute to sustaining or modestly augmenting measured B/Yamagata signals even after removal of the B/Yamagata component [42,43]. Accordingly, any changes in B/Yamagata HAI titers in 2024–2025 as reflecting persistence/back-boosting rather than direct induction by the B/Victoria vaccine component. Vaccines used during the 2020–2021 and 2021–2022 influenza seasons included B/WA/19 (V1A.1) as the B/Victoria vaccine component. During the 3 seasons analyzed in this study, B/AS/21 (clade V1A.3a.2) was the B/Victoria component. Participants vaccinated during these three seasons had strong HAI titers to B/AS/21, but weaker responses to the B/WA/19 strains.

Participants aged 18–79 years vaccinated with Flublok had the strongest serum HAI activity following vaccination (Fig 8), which is consistent with previous studies [44,45]. The Flublok contains the recombinant HA protein, and thereby avoids egg-adaptive antigen changes that may cause antigenic mismatches between the Fluzone vaccines and circulating strains [46]. Elderly participants received either Fluad or Fluzone HD. These participants had significant increases in HAI titers from low baseline D0 HAI titers. However, titers generally remained near or below 1:40 HAI titer against the H1N1 and H3N2 components. A recent randomized trial in people 65 years or older reported no major differences in seroprotection

                                                                                 

rates between Fluad and Fluzone HD, although higher post-vaccination HAI titers were observed with Fluzone HD compared to Fluad during the 2017–2019 seasons [47]. The MF59 adjuvant in Fluad stimulates robust germinal center activity and T helper cells increasing the magnitude, diversity, and affinity of antibodies and may enhance cellular immune responses [48]. Additionally, the Fluzone HD vaccine contains four times the HA per strain (60 μg) compared to the Fluad vaccine (15 μg) [48]. In our cohort, only a few participants (n = 8) aged ≥65 years received Flublok, yielding insufficient sample size for robust statistical comparison; moreover, the substantial age difference between cohorts (mean age ~ 44 years for Flublok vs. ~ 73–74 years for Fluad/Fluzone HD) confounds any indirect comparison.

It is important to note that HAI assays in this study were performed using egg-propagated indicator viruses. This methodology may introduce a bias favoring egg-based vaccines (Fluzone SD/HD, Fluad), because antibodies induced by egg-adapted vaccine antigens can bind more efficiently to egg-grown assay viruses than antibodies induced by cell-based or recombinant vaccines. Despite this potential bias, Flublok elicited significantly higher HAI titers than egg-based comparators, and Flucelvax performed comparably to egg-based vaccines, suggesting that the immunogenicity of these non-egg-based platforms is at least as strong as, and for Flublok likely greater than, that of egg-derived vaccines and that our estimates of their relative advantage are conservative.

Participants vaccinated with Flumist had serum HAI titers that remained unchanged post-vaccination across the three seasons. Flumist vaccine is live attenuated vaccine and administered intranasally. Flumist is designed to induce mucosal immunity, so serum HAI titers might be an incomplete measure of the protective effect of the vaccine. Children vaccinated with Flumist have low HAI seroconversion frequencies following vaccination, but generate local IgA and cellular responses in the respiratory tract [49]. Vaccine effectiveness is comparable between inactivated and live attenuated vaccines among the pediatric population and the live attenuated vaccine was the most effective vaccine against influenza infection (all strains) during the 2019/2020–2022/2023 influenza seasons [50]. Notably, pediatric age span (birth to 8 years) was not included to our study, as younger children often exhibit distinct immune responses to influenza vaccination due to differences in immune maturation, lower baseline immunity, and potentially higher reactogenicity.

Serum samples from vaccinated participants were analyzed by age at D0 and D28 to determine how many times each participant maintained a protective HAI titer (≥1:40) over six timepoints (D0 and D28 of each of three seasons). There was an age-dependent response against the two influenza A viruses. Youngest participants maintain seroprotective HAI (≥1:40) titers while elderly were less likely to maintain HAI titers from season to season. In contrast, participants in all age groups had higher HAI titers against the two influenza B vaccine components compared to HAI titers against influenza A vaccine components. In older adults, these patterns likely reflect historical exposures resulting in D0 baseline cross-reactive antibodies to B/Victoria vaccine strains, while younger participants depend more on recent vaccinations and therefore have lower HAI titers that decline below seroprotective levels by the next season. Overall, repeated vaccination may sustain long-term durable memory B cells and serum seroprotection for B/Yamagata, even though this component was not part of the 2024–2025 influenza vaccine season. However, gaps remain, particularly for B/Victoria in the young, supporting the need for updated formulations and/or booster strategies to maintain seroprotection or HAI titers ≥40.

To address the poorer responses in the elderly, two enhanced vaccines have been widely used for vaccination of people 65 years or older. One is Fluzone High-Dose (HD) containing four-fold higher doses of HA per strain and Fluad that contains a standard HA dose of inactivated influenza virus vaccine with squalene oil-in-water adjuvant, MF59. Fluzone HD and Fluad formulations consistently elicited robust HAI immune responses in elderly participants against the H1N1, H3N2, and influenza B components in the vaccine (Figs 8 and 9). The advantage of Fluzone HD is the elicitation of higher antibody levels, whereas the adjuvanted vaccine can broaden the immune response qualitatively by utilizing natural killer cell activation and T cell responses [48]. However, the Fluzone HD and Fluad adjuvanted vaccines elicit similar seroconversion rates in elderly participants [47].

Participants under 65 years of age were administered one of four influenza vaccines. Overall, Fluzone SD, Flucelvax, and Flublok vaccines all boosted serum HAI activity across three seasons following intramuscular vaccination.

Participants vaccinated with recombinant Flublok vaccine had significantly higher HAI titer increases, especially in participants in the 18–34 and 35–49 years of age. Participants vaccinated with live attenuated vaccine, Flumist, had weaker serum HAI titers across all ages and seasons with no significant rise in serum HAI titers following vaccination.

There are limitations in this study. It was observational in design and relied solely on HAI titers, which may underestimate immune responses to live-attenuated vaccines, such as Flumist that primarily induce mucosal IgA and T-cell immunity. A major limitation is age imbalance between vaccine groups. Age distributions were not comparable between Flublok and Fluzone High-Dose recipients. Fluzone HD was administered almost exclusively to participants aged ≥65 years (mean age~73–74 years), whereas Flublok recipients were predominantly younger adults (mean age~43–45 years). This age imbalance represents a limitation for direct between-vaccine comparisons, as age is a major determinant of HAI responses. In addition, participants were not randomized to vaccine type, and clinical outcomes were not assessed; therefore, the analyses support immunogenicity comparisons and associations, but not causal conclusions about comparative effectiveness. We also did not assess neuraminidase-inhibiting antibodies, cellular responses, or direct vaccine effectiveness against infection, all of which contribute to a fuller understanding of protection. Recent studies have addressed B and T cell responses from this cohort [51,52], but more studies are needed to obtain a comprehensive overview of the impact of each vaccine on the immune system in each age group.

Compared to our previous study that assessed only split-inactivated Fluzone (SD and HD, 2016–2022) [12], the present analysis directly evaluates the HAI responses of recombinant, cell-based, and MF59-adjuvanted vaccines in both adult and elderly populations. Consistent with earlier work, baseline HAI titers declined with age and were lower at the next season's baseline (D0), often leaving older adults below the seroprotection threshold (HAI < 1:40) at the start of the next influenza season. Enhanced formulations (Fluzone HD or Fluad) improve responses in elderly participants [53]. Flublok induced higher HAI responses than Fluzone SD in younger adults, especially against influenza A viruses. HAI-based measures of seroconversion and fold-rise depend heavily on D0 baseline titers and may not directly translate to immune protection against influenza virus infection. Taken together, the results presented in this report support the hypothesis that age, vaccine platform, repeated vaccination, and vaccine updates jointly shape the magnitude and breadth of HAI responses.

## Supporting information

**S1 Table. Vaccine type and number of vaccine recipients during 2022–2023, 2023–2024, and the 2024–2025 influenza seasons.**
(DOCX)

**S2 Table. Source data for Table 1.**
(XLSX)

**S3 Table. Source data for Table 3 and S1 Table.**
(XLSX)

**S4 Table. Source data for Fig 1.**
(XLSX)

**S5 Table. Source data for Figs 2, 3, and 8.**
(XLSX)

**S6 Table. Source data for Table 4 and Figs 4 and 5.**
(XLSX)

**S7 Table. Source data for Figs 6 and 7.**
(XLSX)

**S8 Table. Source data for Fig 9.**
(XLSX)

## Acknowledgments

The authors acknowledge Zachary McGuire, Maya McCoy, Wayne Grant, Spencer Pierce and Michael Carlock, for technical assistance. In addition, the authors would also like to acknowledge UGA FluVac and Cleveland Clinic Florida clinical teams for sample collections blood and saliva processing and technical assistance including, Julia Aguirre, Brittany Baker, Erin Jarrett, Hana Ji, Frankie Stewart, Terris Wimbs, and Emma Whitesell. We would also like to thank all participants enrolled in the study. Some of the influenza viruses were obtained through the Influenza Reagent Resource (IRR), Influenza Division, WHO Collaborating Center for Surveillance, Epidemiology, and Control of Influenza, Centers for Disease Control and Prevention (Atlanta, GA, USA).

## Author contributions

**Conceptualization:** Ted M. Ross.

**Data curation:** Engin Berber.

**Formal analysis:** Engin Berber.

**Funding acquisition:** Ted M. Ross.

**Investigation:** Hannah B. Hanley, Brianna M. Gamez.

**Project administration:** Hannah B. Hanley, Brianna M. Gamez, Ted M. Ross.

**Supervision:** Ted M. Ross.

**Visualization:** Engin Berber.

**Writing – original draft:** Engin Berber.

**Writing – review & editing:** Hannah B. Hanley, Brianna M. Gamez, Ted M. Ross.

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
