## [Decision Letter · Decision Letter 0]

26 Jan 2026

Dear Dr. Ross,

Thank you for submitting your manuscript to PLOS ONE. After careful consideration, we feel that it has merit but does not fully meet PLOS ONE’s publication criteria as it currently stands. Therefore, we invite you to submit a revised version of the manuscript that addresses the points raised during the review process.

**In general the data are well presented, however additional clarification and/or minor corrections are needed to make the paper clearer.  In addition, please consider how to focus the description and discussion on major questions that can be answered by this very comprehensive study.**

We look forward to receiving your revised manuscript.

Kind regards,

Patricia Evelyn Fast, MD, Ph.D.

Academic Editor

PLOS One

**Journal Requirements:**

1. When submitting your revision, we need you to address these additional requirements. Please ensure that your manuscript meets PLOS ONE's style requirements, including those for file naming. The PLOS ONE style templates can be found at https://journals.plos.org/plosone/s/file?id=wjVg/PLOSOne_formatting_sample_main_body.pdf and https://journals.plos.org/plosone/s/file?id=ba62/PLOSOne_formatting_sample_title_authors_affiliations.pdf 2. We note that the grant information you provided in the ‘Funding Information’ and ‘Financial Disclosure’ sections do not match.  When you resubmit, please ensure that you provide the correct grant numbers for the awards you received for your study in the ‘Funding Information’ section. 3. Thank you for stating the following financial disclosure: This project has been funded by the National Institute of Allergy and Infectious Diseases, a component of the NIH, Department of Health and Human Services, under contract 75N93019C00052   Please state what role the funders took in the study.  If the funders had no role, please state: "The funders had no role in study design, data collection and analysis, decision to publish, or preparation of the manuscript." If this statement is not correct you must amend it as needed. Please include this amended Role of Funder statement in your cover letter; we will change the online submission form on your behalf. 4. When completing the data availability statement of the submission form, you indicated that you will make your data available on acceptance. We strongly recommend all authors decide on a data sharing plan before acceptance, as the process can be lengthy and hold up publication timelines. Please note that, though access restrictions are acceptable now, your entire data will need to be made freely accessible if your manuscript is accepted for publication. This policy applies to all data except where public deposition would breach compliance with the protocol approved by your research ethics board. If you are unable to adhere to our open data policy, please kindly revise your statement to explain your reasoning and we will seek the editor's input on an exemption. Please be assured that, once you have provided your new statement, the assessment of your exemption will not hold up the peer review process. 5. Please upload a new copy of Figures 1 – 9, as the detail is not clear. Please follow the link for more information:  https://journals.plos.org/plosone/s/figures 6. If the reviewer comments include a recommendation to cite specific previously published works, please review and evaluate these publications to determine whether they are relevant and should be cited. There is no requirement to cite these works unless the editor has indicated otherwise. 

**Additional Editor Comments:**

Please find a number of comments to your paper, almost all of which are aimed to improve its clarity. Please carefully review and respond to these.

Major comments:

It’s difficult for an observational study with a huge amount of data but try to emphasize what the questions were that the study hoped to answer and whether it could answer them. For example, how do potential vaccines for elderly compare in antibody generating capacity? Limitation: Flublok not used in elderly.

Example: what is the effect of skipping years? There might be better responses after a skipped year, but also more vulnerability during that skipped year. (recognizing that a study such as this could not provide sufficient data to call into question a standing recommendation.)

It seems that the data describing the tables and graphs is in the text rather than in Legends/Footnotes. Each table and graph should be able to stand alone and be interpretable. Sometimes, they are not clearly enough labelled to do so.

Please clarify as much as possible in Table/Figure and text when data relate to the repeater cohort and when to the overall population.

Minor comments

Abstract:

Clarify whether the lower titers in elderly people were in fact a factor in the calculated higher seroconversion rates, i.e. by allowing individuals to regain seronegative status prior to the next year’s vaccination, were more seroconversions (which require seronegativity??), made possible?

For the Flublok vs Hi dose FluZone comparison, state whether ages were comparable in the 2 groups. An age disparity would be a limitation on comparing the vaccines.

Line 46-8 “…serum HAI activity was similar in elderly participants 47 following vaccination with either Fluzone HD or Fluad against both influenza A or B vaccine 48 components”. Please clarify what is being compared here; is it FluA and FluB titers after the two vaccines mentioned? Or is there a broader comparison here?

Methods. No vaccine in the 4 weeks since last vaccination seems an unusual criterion, since people of this age would not be expected to receive two influenza vaccines only 4 weeks apart. Is there some rationale that could be explained?

Table 2 Footnote. IAV (H1N1) strains are selected to reflect circulating (missing word?),

Table 3. If an individual in the repeater cohort moved from one age category to the next over 3 years, how were the data handled? Is Table 3 the original age or is the actual age in that year used?

Line 210. In discussion, consider effect of overall greater age?

Line 212. The proportion of adolescents (≤17 years) was relatively small in the 2022-2023 season (9.5%) 213 and 2024-2025 season (5.3%) but increased to 6.5% in 2023-2024 season. The text seems to imply the last season was greater than the first, but in fact first season is greatest. Consider rewording?

Line 239. Consider stating ages receiving FluMist, and whether recipients had previously been vaccinated with IAV. Prior HAI titer might interfere with take.

Question: were all vaccinations given according to manufacturers age recommendations/indications?

FIGURE 1

Needs a legend, which I did not find. Is it correct that the first column is D0 and the second D28 under each strain?

FIGURE 2

It’s not easy to follow the rise in titer within each age group for each year. The lines connect baseline to baseline and post-vaccination to post-vaccination, which does not seem like the primary focus. Perhaps all values could be on a 0, 28 day timeline with before and after vaccination linked by a line. Then the confidence intervals could be displayed.

Line 242. “Table in S1 Table summarizes” wording is awkward

Line 270 “had similar baseline HAI 270 titers from participants” perhaps say similar TO?

Line 263. The heat map is a good idea. Shouldn’t this explanation be, at least in part, in the legend? What does the blue column on th right (H3N2) mean? It appears to mean participants are ordered by decreasing titer, but they don’t seem to be.

LINE 285. “ Fewer participants between 18–34 years old had HAI titers >1:40 at D0 during the 2022-2023 season compared the 287 next two seasons, but had HAI titers >1:40 following vaccination on D28, with modest back288 boosting against historical H3N2 isolates. The point of the second part of the sentence is not clear—is a word missing??

Line 480 and following and Table 4. Is this referring only to the final season of the study? How do 2017-2019 come into it (footnote to Table 4)? Line 485…which participants are meant by “these participants”? Line 490 and following is very hard to follow. Should it say …participants WHO were not……? Is the B Victoria component of the vaccine influencing the B Yamagata response? Please comment in Discussion section.

Line 530… those who switched from SD to…word(s) missing ???

It’s hard at first to see what’s different between two paragraphs in results describing Figs 6 and 7, and the two figures. Titles emphasizing IAV and IBV would help.

Line 668-671. ….’a smaller proportion had detectable titers and achieved…. “ Possibly this could be more clearly stated as ‘ a smaller proportion fell into the group that initially had detectable titers and then achieved..”

Line 714. A D amino acid sounds like chirality is being discussed. ‘..had the amino acid D’ might be clearer.

Is it correct that titers to B Yamagata rose during the 2024-25 season even though it was not in the vaccine? Could this be because of a few outliers who got infected with B Yamagata and raised the average or extreme cross-boosting?

Line 737 ‘and thereby avoidS…

Line 738… while this statement (may avoid mismatches with circulating strains due to egg adaptation) may be correct, does it explain your results? Methods states you grew the indicator viruses for the HAI assays in eggs? (Was that a disadvantage for FluBlok and FluCelVax measured titers? Please comment.

735 and following—please clarify whether your data sheds any light on the relative value of Flublok vs FluAd and/or Fluzone HD in older adults.

748 and following—it might be worth looking at titers in pediatric participants alone. Also, perhaps mention that half the age span of pediatric patients in general was omitted from this study. Younger children’s responses probably differ.

Line 762. “In contrast, participants in all age groups had (descriptive word missing?) HAI titers against the two influenza B vaccine components compared to HAI titers against influenza A vaccine components.”

Reviewers' comments:

**Comments to the Author**

1. Is the manuscript technically sound, and do the data support the conclusions?

Reviewer #1: Yes

2. Has the statistical analysis been performed appropriately and rigorously?

Reviewer #1: I Don't Know

3. Have the authors made all data underlying the findings in their manuscript fully available?

Reviewer #1: Yes

4. Is the manuscript presented in an intelligible fashion and written in standard English?

Reviewer #1: Yes

**Reviewer #1:**  Reviewer Comments: Reviewer Comments:

Manuscript Number: PONE-D-25-48769

Title: Assessment of Hemagglutinin-Inhibition Activity Following Influenza Vaccination During the 2022-2023, 2023-2024, and 2024-2025 Seasons.

This is a 3-season report on the immunogenicity of six influenza vaccine types in 1328 participants aged 9-89 years old enrolled in an observational cohort study, 203 of whom received an influenza vaccine in all 3 seasons allowing longitudinal assessment. Overall, the manuscript is well written, however there are a few major and multiple minor revisions needed.

Major Comments:

1. Page 21, Lines 165-167: HAI titers of ≥40 are considered non-detectable or seronegative in this manuscript, which is incorrect as the lower limit of detection of HAI test is a dilution of 1:10, and <10 is considered not detectable or seronegative. Also, a titer of ≥40 or more is considered seropositive or seroprotection (if it is a post-vaccination titer), but seroprotection and seropositivity are used interchangeably on D0 in the figures. Consider using alternative terminology such as no/low HAI titer <40 vs. HAI titer ≥40. Other alternatives are using no-seroprotection vs. seroprotection. The definition of seroconversion looks good.

2. Page 22, Line 195: Regarding comparisons between vaccine types, Dunnett’s multiple comparison test is appropriate if there is one control group and the one-way ANOVA is significant. If there are multiple comparison groups, then Tukey or Scheffe test may be more appropriate.

3. Page 24, Line 245: Supplemental Table 1 and text have discrepant vaccine types – e.g. line 245 (FB>FB>FB; n=16) whereas Supplemental Table 1 shows FM>FM>FM, n=16 and does not show anyone with FB>FB>FB. Also, this Table shows FB > FC > FB = 8 in row 6 from top and again FB > FC > FB = 1 four rows up from the bottom Total row. Similarly, FZ SD >FZ HD > FZ HD are listed twice with N = 5 and 1 each.

Minor Comments:

The manuscript needs correction or clarification at multiple places including in the tables and figures.

1. Page 11, Line 59: Please clarify if the lower vaccine-induced response is meant for standard-dose inactivated vaccine.

2. Page 12, Line 89: Please clarify if individual vaccinees with more than one dose in a study season were excluded or not.

3. Page 19, Line 121, Table 3: Please clarify with a footnote when the age group and BMI are for individuals in Table 3. ?First vaccination, last vaccination or some other time.

4. Page 21, Line 173. Please spell out what BEI stands for.

5. Page 23, Lines 212-213: The proportion of adolescents (≤17 years) was relatively small in the 2022-2023 season (9.5%) and 2024-2025 season (5.3%) but increased to 6.5% in 2023-2024 season. Please correct this sentence because the proportion of children <18 years decreased from 9.5% to 6.5% to 5.3% in the 3 consecutive study seasons.

6. Page 23, Lines 216-218: Consider deleting this sentence here because supporting data is shown later under vaccination patterns in repeaters.

7. Page 25, line 267: Figure 1A Y-axis label shows ages 10-17 years instead of 9-17 years.

8. Page 25, line 276: There is no titer done at 6 months from the pre-vaccination baseline so it might be better to say that the titer had waned prior to vaccination next season rather than wanes during the season.

9. Page 25, lines 283- 285: There were no titers done 365 days after 2024-25 season and DR21 titers seem to have waned by 2024-25 baseline, please clarify this sentence.

10. Page 28, line 361: Statistically “similar” or “significant”?

11. Page 28, line 378: Clarify seroconverted against A (H1N1).

12. Page 28, line 384: Is the 28% B/Yamagata seroconversion correct for average over 2 seasons and is it truly the lowest? Is it lower than 31% average seroconversion for A (H1N1) over 3 seasons because average B/Yamagata seroconversion needs to be assessed over first 2 seasons only.

13. Page 29, Figure 4, line 401. There are no numbers seen on the X-axis in this figure, it shows the seasons.

14. Page 30, line 418: Re ages 18-34 years, should this be 35-49 years?

15. Page 30, line 420: Re 33-53% against A (H3N2)? Please clarify.

16. Page 30, lines 428-429, re B/Vic: Oldest participants show higher seroconversion than youngest in 2 of 3 seasons.

17. Page 30, lines 434-435: Number of participants not seen along the X-axis in Fig 5.

18. Page 30, lines 434-435: Consider replacing UGA9 with 2024-2025 season.

19. Page 32, Table 4, line 478: Seasons listed need corrections.

20. Page 32, Table 4, line 479: It is not clear if the Fisher’s exact test was used for this table as results are not shown or described in the text until the discussion – please move this to the results or table.

21. Page 33, line 493: Correction: 30% did seroconvert. Delete not at the end of the sentence.

22. Page 35, line 556, consider naming the six vaccines here.

23. Page 36, line 588: Correction - 2023-2024 season (Fig 9B) compared to 202-23 season (Fig 9A).

24. Page 37, Fig 9, line 619: Red Asterix is not seen.

25. Page 38, line 645 – correction, to historical strains rather than to drifted viral variants.

26. Page 39, line 662, consider saying to previous two years rather than 1-2 years.

27. Page 39, line 669, change detectable to protective or whatever alternative terminology is used, as appropriate.

28. Page 41, line 735, specify participants age range for Flublock

29. Page 42, line 770: to maintain ‘seroprotection or HAI titers ≥40’.

**Do you want your identity to be public for this peer review?** For information about this choice, including consent withdrawal, please see our For information about this choice, including consent withdrawal, please see our Privacy Policy .

Reviewer #1: No

---

## [Author Response · Author response to Decision Letter 1]

13 Mar 2026

March 8, 2026

Patricia Evelyn Fast, MD, Ph.D.

Academic Editor

PLOS One

Response to Reviewers and Academic Editor

Manuscript ID: PONE-D-25-48769

Title: Assessment of Hemagglutinin-Inhibition Activity Following Influenza Vaccination During the 2022-2023, 2023-2024, and 2024-2025 Seasons

Journal: PLOS ONE

Journal Requirements: 1.

Author response: We formatted the title page and main text to match the PLOS ONE style templates, and ensure all submission file names follow PLOS ONE conventions (e.g., Manuscript, Revised Manuscript with Track Changes, Response to Reviewers, Figures, Supporting Information).

Journal Requirements: 2.

We note that the grant information you provided in the ‘Funding Information’ and ‘Financial Disclosure’ sections do not match.

Author response: We reconciled and made the grant/contract numbers consistent across (i) the manuscript Funding section and (ii) the online submission ‘Funding Information’ and ‘Financial Disclosure’ fields.

Journal Requirements: 3.

Thank you for stating the following financial disclosure:

This project has been funded by the National Institute of Allergy and Infectious Diseases, a component of the NIH, Department of Health and Human Services, under contract 75N93019C00052

Author response: We confirm that “The funders had no role in study design, data collection and analysis, decision to publish, or preparation of the manuscript.” Please do not hesitate to update the online submission fields on our behalf to reflect the following funding information:

TMR: Contract 75N93019C00052; National Institute of Allergy and Infectious Diseases (NIAID), National Institutes of Health (NIH), U.S. Department of Health and Human Services; https://www.niaid.nih.gov/ and https://www.nih.gov/

TMR: GRA-001; Georgia Research Alliance; https://gra.org/

During submission:

In the manuscript draft:

Journal Requirements: 4.

When completing the data availability statement of the submission form, you indicated that you will make your data available on acceptance. We strongly recommend all authors decide on a data sharing plan before acceptance, as the process can be lengthy and hold up publication timelines. Please note that, though access restrictions are acceptable now, your entire data will need to be made freely accessible if your manuscript is accepted for publication. This policy applies to all data except where public deposition would breach compliance with the protocol approved by your research ethics board. If you are unable to adhere to our open data policy, please kindly revise your statement to explain your reasoning and we will seek the editor's input on an exemption. Please be assured that, once you have provided your new statement, the assessment of your exemption will not hold up the peer review process.

Author response: Thanks for your attention. Now, we made all data underlying the findings are provided in the Supporting Information files (S1 Data–S8 Data).

Journal Requirements: 5.

Please upload a new copy of Figures 1 – 9, as the detail is not clear. Please follow the link for more information: https://journals.plos.org/plosone/s/figures

Author response: We provided Figures 1–9 at publication-quality resolution and confirmed that text, axes, and symbols remain legible at final size, consistent with PLOS ONE figure specifications.

Journal Requirements: 6.

Author response: No specific citation recommendations were provided by the reviewer in the comments received. We nonetheless reviewed the relevant literature and added citations where they directly support or contextualize our findings.

Additional Editor Comments:

Please find a number of comments to your paper, almost all of which are aimed to improve its clarity. Please carefully review and respond to these.

Dear Editor.

Thank you for your careful review and for the constructive comments provided on our manuscript. We appreciate the editor’s and reviewers’ efforts to improve the clarity and presentation of this work. We have carefully revised the manuscript throughout in response to these comments. Specifically, we clarified the primary study questions and the scope of inference appropriate for this observational study, expanded and revised figure legends and table footnotes so they can stand alone, corrected and refined the relevant Results and Discussion text, and addressed points of wording, interpretation, and methodological clarity raised in the review.

In addition, during preparation of the revised manuscript, we carefully re-verified the dataset used to generate Table 4 against the original Excel source files. During this verification step, we identified discrepancies that arose during manual transfer of values from the analysis spreadsheet into the manuscript table. To ensure complete accuracy, we rechecked the entire dataset and regenerated Table 4 directly from the verified data. The table has now been corrected, and the corresponding text in the Results and Discussion sections has been revised to ensure consistency with the updated values.

These revisions do not affect the overall conclusions of the study but improve the accuracy and clarity of the reported results. We appreciate the opportunity to carefully review and correct this section during the revision process. All relevant data used in the tables and figures have been supplemented.

Major comment:

It’s difficult for an observational study with a huge amount of data but try to emphasize what the questions were that the study hoped to answer and whether it could answer them. For example, how do potential vaccines for elderly compare in antibody generating capacity? Limitation: Flublok not used in elderly.

Example: what is the effect of skipping years? There might be better responses after a skipped year, but also more vulnerability during that skipped year. (recognizing that a study such as this could not provide sufficient data to call into question a standing recommendation.)

Author response: We have revised the manuscript to explicitly state the primary study questions in the end of Introduction and refined the Discussion around these questions addressing these questions.

We also added a clear limitation noting that Flublok was not administered to participants ≥65 years in this cohort, which constrains comparative conclusions in older adults. Finally, we clarified what our observational data can and cannot infer regarding “skipped years”: we describe associations between prior-season vaccination history and baseline titers and post-vaccination fold-rise, but the study design does not quantify the clinical tradeoff between vulnerability during a gap year and any potential boosting after a vaccination gap, and it does not support changes to vaccination recommendations.

Please see Lines 89–95 (study objectives in Introduction)

Lines 782–787 (discussion of skipped vaccination years)

Lines 930–939 (limitations regarding Flublok use in elderly)

Major comment:

It seems that the data describing the tables and graphs is in the text rather than in Legends/Footnotes. Each table and graph should be able to stand alone and be interpretable. Sometimes, they are not clearly enough labelled to do so.

Please clarify as much as possible in Table/Figure and text when data relate to the repeater cohort and when to the overall population.

Author response: We have revisited all figure and table legends to ensure that each figure/table is interpretable without relying on the main text. We moved essential definitions into the legends/footnotes, including D0 vs D28, the seroprotection threshold, the seroconversion definition, cohort definitions, sample sizes, and statistical tests. We also clarified, for each figure/table, whether the data are derived from the overall seasonal cohorts or the longitudinal repeater cohort.

Additional Editor Comments: Minor comments

Minor comment (Abstract):

Abstract:

Clarify whether the lower titers in elderly people were in fact a factor in the calculated higher seroconversion rates, i.e. by allowing individuals to regain seronegative status prior to the next year’s vaccination, were more seroconversions (which require seronegativity??), made possible?

Author response: We made it clear in the Abstract that higher seroconversion rates in older adults can be driven by lower baseline (D0) titers, because the seroconversion definition depends on a ≥4-fold rise and a post-vaccination threshold. We also avoid implying that “seronegativity” is required beyond the formal definition used. Lines 42-45.

Minor comment:

For the Flublok vs Hi dose FluZone comparison, state whether ages were comparable in the 2 groups. An age disparity would be a limitation on comparing the vaccines.

Author response: Age distributions were not comparable between Flublok and Fluzone High-Dose recipients. Fluzone HD was administered almost exclusively to participants aged ≥65 years (mean age ~73–74 years), whereas Flublok recipients were predominantly younger adults (mean age ~43–45 years). This age imbalance represents a limitation for direct between-vaccine comparisons, as age is a major determinant of HAI responses. We set this major limitation of this study and have been added to the discussion. Lines 930-938.

Minor comment (Abstract):

Line 46-8 “…serum HAI activity was similar in elderly participants 47 following vaccination with either Fluzone HD or Fluad against both influenza A or B vaccine 48 components”. Please clarify what is being compared here; is it FluA and FluB titers after the two vaccines mentioned? Or is there a broader comparison here?

Author response: We revised the sentence to clarify that the comparison is specifically between Fluzone HD and Fluad recipients in elderly participants, and that post-vaccination HAI titers are compared for each vaccine component (influenza A and influenza B strains). These results can be found in Figs 8 and 9. Please see lines 48-50.

Minor comment (Methods):

No vaccine in the 4 weeks since last vaccination seems an unusual criterion, since people of this age would not be expected to receive two influenza vaccines only 4 weeks apart. Is there some rationale that could be explained?

Author response: We removed the “within 4 weeks” exclusion criterion. Instead, eligibility required that participants had not yet received the seasonal influenza vaccine in that influenza season. We also clarified that concurrent participation in another influenza vaccine study was an exclusion criterion to avoid overlapping vaccinations or study interventions that could confound immunogenicity outcomes. Lines 97-98.

Table 2 Footnote. IAV (H1N1) strains are selected to reflect circulating (missing word?)

Author response: Thank you for noting this. We corrected the Table 2 footnote to include the missing word and to clarify the wording: “….reflect circulating viruses”. 119-125

Minor comment (Table 3):

If an individual in the repeater cohort moved from one age category to the next over 3 years, how were the data handled? Is Table 3 the original age or is the actual age in that year used?

Author response: Table 3 summarizes demographics for the longitudinal repeater cohort at enrollment (baseline) for descriptive purposes. We have added “Age and BMI represent measurements at first enrollment (2022–2023 season) for the longitudinal repeater cohort” to the Table 3 footnote. Line 144.

Minor comment (Discussion):

Line 210. In discussion, consider effect of overall greater age?

Author response: We have added a new paragraph in the Discussion (lines 723-729) to explicitly address the demographic confounding introduced by the increasing mean age of our enrolled population across the three seasons (from 47.1 years to 52.4 years in 2024–2025). Lines 720-733.

Minor comment:

Line 212. The proportion of adolescents (≤17 years) was relatively small in the 2022-2023 season (9.5%) 213 and 2024-2025 season (5.3%) but increased to 6.5% in 2023-2024 season. The text seems to imply the last season was greater than the first, but in fact first season is greatest. Consider rewording?

Author response: Thank you for pointing this out. We revised the text to clarify the temporal trend across seasons and to avoid any unintended implication that the later season exceeded the earlier one. The revised sentence now states that the proportion of adolescents (≤17 years) decreased from 9.5% in 2022–2023 to 6.5% in 2023–2024 and further to 5.3% in 2024–2025. Lines 240-243.

Minor comment:

Line 239. Consider stating ages receiving FluMist, and whether recipients had previously been vaccinated with IAV. Prior HAI titer might interfere with take.

Author response: FluMist recipients were primarily younger participants, consistent with the licensed age indication (2–49 years). Line 267-269.

Prior HAI titers from prior IAV vaccination may interfere with LAIV “take” by limiting vaccine virus replication and shedding in primed individuals, which could contribute to the low measurable serum HAI responses observed after LAIV administration due to neutralization. In our repeater cohort, FluMist was administered to 16 participants across all three seasons (FM>FM>FM).

Within the study period, only three repeaters received Flublok (recombinant IAV) in UGA7 followed by FluMist in UGA8 and UGA9 (FB>FM>FM). Because detailed influenza vaccination history prior to study enrollment was not consistently available for all FluMist recipients, we were unable to comprehensively stratify FluMist outcomes by prior IAV vaccination status beyond the within-study repeat-vaccination patterns described.

Minor comment (Question):

Question: were all vaccinations given according to manufacturers age recommendations/indications?

Author response: All vaccines were administered according to licensed manufacturer age indications and FDA-approved, age-based recommendations, as reflected by the age distributions for each vaccine type in our cohort (FluMist: 2–49 years; Fluzone SD/HD: ≥6 months/≥65 years; Flucelvax: ≥6 months; Flublok: ≥18 years; Fluad: ≥65 years). We have included this statement: “All vaccines were administered according to licensed manufacturer age indications and FDA-approved, age-based recommendations.” in our revised draft: 244-246.

Minor comment (FIGURE 1):

Needs a legend, which I did not find. Is it correct that the first column is D0 and the second D28 under each strain?

Author response: Figure legends are placed after the first paragraph where the figure is cited. We have revised figure legends with details as requested, and have stated “Within each strain, paired columns represent D0 (left) and D28 (right)”. Line 284-301.

Minor comment (FIGURE 2):

It’s not easy to follow the rise in titer within each age group for each year. The lines connect baseline to baseline and post-vaccination to post-vaccination, which does not seem like the primary focus. Pe

---

## [Editor Report · Decision Letter 1]

1 Apr 2026

Assessment of Hemagglutinin-Inhibition Activity Following Influenza Vaccination During the 2022-2023, 2023-2024, and 2024-2025 Seasons

PONE-D-25-48769R1

Dear Dr. Ross:

We’re pleased to inform you that your manuscript has been judged scientifically suitable for publication and will be formally accepted for publication once it meets all outstanding technical requirements.

Kind regards,

Patricia Evelyn Fast, MD, Ph.D.

Academic Editor

PLOS One

Additional Editor Comments (optional):

Thank you for your careful attention to the reviews.
---

## [Editor Report · Acceptance letter]

PONE-D-25-48769R1

PLOS One

Dear Dr. Ross,

I'm pleased to inform you that your manuscript has been deemed suitable for publication in PLOS One. Congratulations! Your manuscript is now being handed over to our production team.

Kind regards,

on behalf of

Dr. Patricia Evelyn Fast

Academic Editor

PLOS One